HYPOTHESIS

Voltage-Gated Na Channels

# Isoform-specific N-linked glycosylation of Na$_V$ channel α-subunits alters β-subunit binding sites

Christopher A. Beaudoin[1], Manas Kohli[1], Samantha C. Salvage[1], Hengrui Liu[1], Samuel J. Arundel[1], Samir W. Hamaia[1], Ming Lei[2], Christopher L.-H. Huang[1,3], and Antony P. Jackson[1]

Voltage-gated sodium channel α-subunits (Na$_V$1.1–1.9) initiate and propagate action potentials in neurons and myocytes. The Na$_V$ β-subunits (β1–4) have been shown to modulate α-subunit properties. Homo-oligomerization of β-subunits on neighboring or opposing plasma membranes has been suggested to facilitate cis or trans interactions, respectively. The interactions between several Na$_V$ channel isoforms and β-subunits have been determined using cryogenic electron microscopy (cryo-EM). Interestingly, the Na$_V$ cryo-EM structures reveal the presence of N-linked glycosylation sites. However, only the first glycan moieties are typically resolved at each site due to the flexibility of mature glycan trees. Thus, existing cryo-EM structures may risk de-emphasizing the structural implications of glycans on the Na$_V$ channels. Herein, molecular modeling and all-atom molecular dynamics simulations were applied to investigate the conformational landscape of N-linked glycans on Na$_V$ channel surfaces. The simulations revealed that negatively charged sialic acid residues of two glycan sites may interact with voltage-sensing domains. Notably, two Na$_V$1.5 isoform-specific glycans extensively cover the α-subunit region that, in other Na$_V$ channel α-subunit isoforms, corresponds to the binding site for the β1- (and likely β3-) subunit immunoglobulin (Ig) domain. Na$_V$1.8 contains a unique N-linked glycosylation site that likely prevents its interaction with the β2 and β4-subunit Ig-domain. These isoform-specific glycans may have evolved to facilitate specific functional interactions, for example, by redirecting β-subunit Ig-domains outward to permit cis or trans supraclustering within specialized cellular compartments such as the cardiomyocyte perinexal space. Further experimental work is necessary to validate these predictions.

## Introduction

Voltage-gated sodium (Na$_V$) channels initiate and propagate the rising phase of the action potential in electrically excitable cells, such as neurons and myocytes (Yu and Catterall, 2003). Na$_V$ channels are transmembrane proteins that consist of an ion-selective α-subunit (molecular mass ~250 kDa) and associated regulatory β-subunits (molecular mass ~30 kDa). There are nine structurally distinct Na$_V$ channel α-subunit isoforms (Na$_V$1.1–1.9), which are relatively tissue-specific: e.g., Na$_V$1.1–1.3 and Na$_V$1.6 are largely found in the central nervous system, Na$_V$1.4 is found in skeletal muscle, Na$_V$1.5 is primarily expressed in cardiac muscle, and Na$_V$1.7–1.9 are considered to be specific to the peripheral nervous system (Clare et al., 2000). However, several exceptions in tissue-specific Na$_V$ channel expression have been discovered. For example, Na$_V$1.8 has been detected in the aging heart (Dybkova et al., 2018) and mutations in Na$_V$1.8

have been associated with cardiac pathologies, although this may be explained by the presence of Na$_V$1.8 in intracardiac neurons (Verkerk et al., 2012). Alternatively spliced Na$_V$ channels add further variations, including an exon-skipping event for Na$_V$1.5 in macrophages (Rahgozar et al., 2013). Further investigation into the sequence- and structure-specific differences in Na$_V$ channels may elucidate their functional roles in different tissues.

The revolution in cryogenic electron microscopy (cryo-EM) has revealed unparalleled insights into the structures of the Na$_V$ channels and the regulatory β-subunits (Jiang et al., 2022; Blundell and Chaplin, 2021; Liang et al., 2022). High-resolution cryo-EM structures of all human Na$_V$ channel α-subunits except Na$_V$1.9 have been published (Noreng et al., 2021). The Na$_V$ channel tertiary structure is comprised of four internally

---

[1]Department of Biochemistry, Hopkins Building, University of Cambridge, Cambridge, UK; [2]Department of Pharmacology, University of Oxford, Oxford, UK; [3]Department of Physiology, Development and Neuroscience, University of Cambridge, Cambridge, UK.

Correspondence to Antony P. Jackson: apj10@cam.ac.uk

M. Kohli's current affiliation is Institute of Cancer Research, London, UK. This work is part of a special issue on Voltage-Gated Sodium (Na$_v$) Channels.

homologous domains (DI–DIV) that enclose a central pore in which ion selectivity is determined by size and charge (Fig. 1, A and B) (Catterall, 2000). The pore is surrounded by extracellular turret loops (ECTLs), contributed by each of the domains (Stephens et al., 2015). Helices S1–S4 of each domain form the voltage-sensing domain (VSD), in which positively charged lysine and arginine residues line one face of the S4 α-helix (Catterall, 1986). In response to changes in membrane potential, the movement of the S4 helices in domains I–III induces channel activation, whilst the subsequent movement of the domain IV S4 helix induces inactivation (Angsutararux et al., 2021).

Na$_V$ β-subunits (β1–4) modulate channel gating, trafficking, and kinetics (Namadurai et al., 2015). The β-subunits comprise a single N-terminal extracellular Ig-domain that is connected via a flexible linker to a transmembrane domain and C-terminal intracellular tail (Cusdin et al., 2010; Salvage et al., 2020a). The interactions between several of the Na$_V$ channels and β-subunits have been determined by cryo-EM. Specifically, human β1 has been resolved with Na$_V$1.1 (Pan et al., 2021), Na$_V$1.3 (Li et al., 2022), Na$_V$1.4 (Pan et al., 2018), Na$_V$1.6 (Fan et al., 2023), and Na$_V$1.7 (Shen et al., 2019; Huang et al., 2022). Human β2 has been resolved with Na$_V$1.3 (Li et al., 2022), Na$_V$1.6 (Fan et al., 2023), and Na$_V$1.7 (Shen et al., 2019; Huang et al., 2022). Human β4 has been resolved with Na$_V$1.1 (Pan et al., 2021). In all of these structures, the β1 transmembrane domain makes extensive contacts with the S1 and S2 helices of the α-subunit domain III VSD, while the β1 Ig-domain contacts extracellular regions in the domain III VSD and the ECTL regions of domains I and IV (Fig. 1, C and D). In contrast, the Ig-domains of β2 and β4 contact the α-subunit domain II ECTL via a covalent disulfide bond, but with no apparent interactions of the β-subunit transmembrane domain. The binding site of β3 is not yet clear. However, the strong sequence similarity between β3 and β1 (Namadurai et al., 2015) suggests that β3 may bind to these Na$_V$ channel isoforms in a similar way as β1 (Zhu et al., 2017). Indeed, the recent cryo-EM structure of β3 and the non-classical sodium channel Na$_X$ is consistent with this view (Noland et al., 2022). The β-subunits are structurally related to members of the Ig-domain family of cell-adhesion molecules. They have the potential to form trans and cis homo-oligomers that interact with β-subunits on opposing membranes or neighboring plasma membranes, respectively, via their Ig-domains: the β1, β2, and β4 have been demonstrated to interact in trans and the β3 subunit has been suggested to interact in cis (Salvage et al., 2020a). Such homophilic interactions may lead to supramolecular clustering between the Na$_V$ channels to modulate localized depolarization (Salvage et al., 2020b).

The extracellular domains of plasma membrane proteins often contain one or more copies of the canonical N-linked glycosylation motif: NX[S or T], where X is any amino acid except proline (Gavel and von Heijne, 1990). Changes in N-linked glycosylation, e.g., as a result of a mutation, may result in changes in protein folding, intra- and intermolecular interactions, and degradation (Freeze and Schachter, 2009; Beaudoin et al., 2022). In the endoplasmic reticulum, a core glycan structure containing two N-acetylglucosamine (GlcNAc) residues and three mannose

residues can be covalently added to the asparagine residue of this motif, provided it is accessible to the glycosylation enzymes (Reily et al., 2019). During further maturation, the outer branches of the tree become heterogeneous, reflecting the stochastic addition and removal of individual sugar residues as the protein traffics through the secretory pathway (Cherepanova et al., 2016). These further modifications may include the addition of one or more sialic acid residues at the terminal positions (Nagae et al., 2020). Na$_V$ channel α-subunits contain multiple N-linked glycosylation sites in their ECTLs. Interestingly, the Na$_V$ channel cryo-EM data all show the presence of additional electron density around the relevant ECTL asparagine residues, indicating that they are indeed glycosylated. However, usually only the primary N-acetyl glucosamine rings, covalently attached to the asparagine residue, are resolved since the full glycan trees are normally too flexible to determine single structures (Atanasova et al., 2020). Thus, the existing cryo-EM structures may risk de-emphasizing the full extent of the glycan trees and their structural implications. Herein, the structural effects of glycans are questioned for Na$_V$1.5 and other Na$_V$ channel isoforms, with particular emphasis on the positioning of sialic acid residues close to VSDs and the potential for steric clashes between β-subunits and the glycosylated α-subunit (Salvage et al., 2020a).

## Results

### Structural modeling and spatial organization of N-linked glycans on Na$_V$ channels

The cryo-EM models of Na$_V$1.1–1.8 are all resolved with at least the first glycan-detectable moiety at various NX[S or T] motifs exposed on the ECTLs. These sites provide experimental references for predictions that may uncover additional potential sites. The preliminary use of sequence-based bioinformatics tools (NetNGlyc [Gupta and Brunak, 2002], N-GlycoSite [Zhang et al., 2004]) to predict N-linked glycosylation resulted in several false-negative and false-positives. Therefore, only those sites where an asparagine-proximal N-acetyl glucosamine residue was discovered on the cryo-EM structures were inspected for their structural impact (Fig. 2 A).

Although mature N-linked glycan trees are heterogeneous, they all share a common core sequence of N-acetylglucosamine, mannose, and galactose. Most are "complex" with branching antennae. The tips typically contain at least one sialic acid residue. Previous work that is currently under peer review has identified the most prevalent glycan tree on the β3-subunit as a core-fucosylated, bi-antennary structure containing one terminal sialic acid. Hence, this form was used as a representative template for glycan modeling (Fig. 2, B and C). Alignment of the Na$_V$ channel primary sequences with reference to the resolved sites on the cryo-EM structures reveals three sites common to all channels (corresponding to N291, N1365, and N1380 on Na$_V$1.5), one site (N328 on Na$_V$1.5) on domain I shared between all channels except Na$_V$1.7 and Na$_V$1.9, and one site (N1388 on Na$_V$1.5) on domain III shared between Na$_V$1.5, Na$_V$1.7, and Na$_V$1.8 (Fig. 3 A). Of note, the glycan moieties of some conserved NX[S or T] motifs were not seen in the cryo-EM structures since these

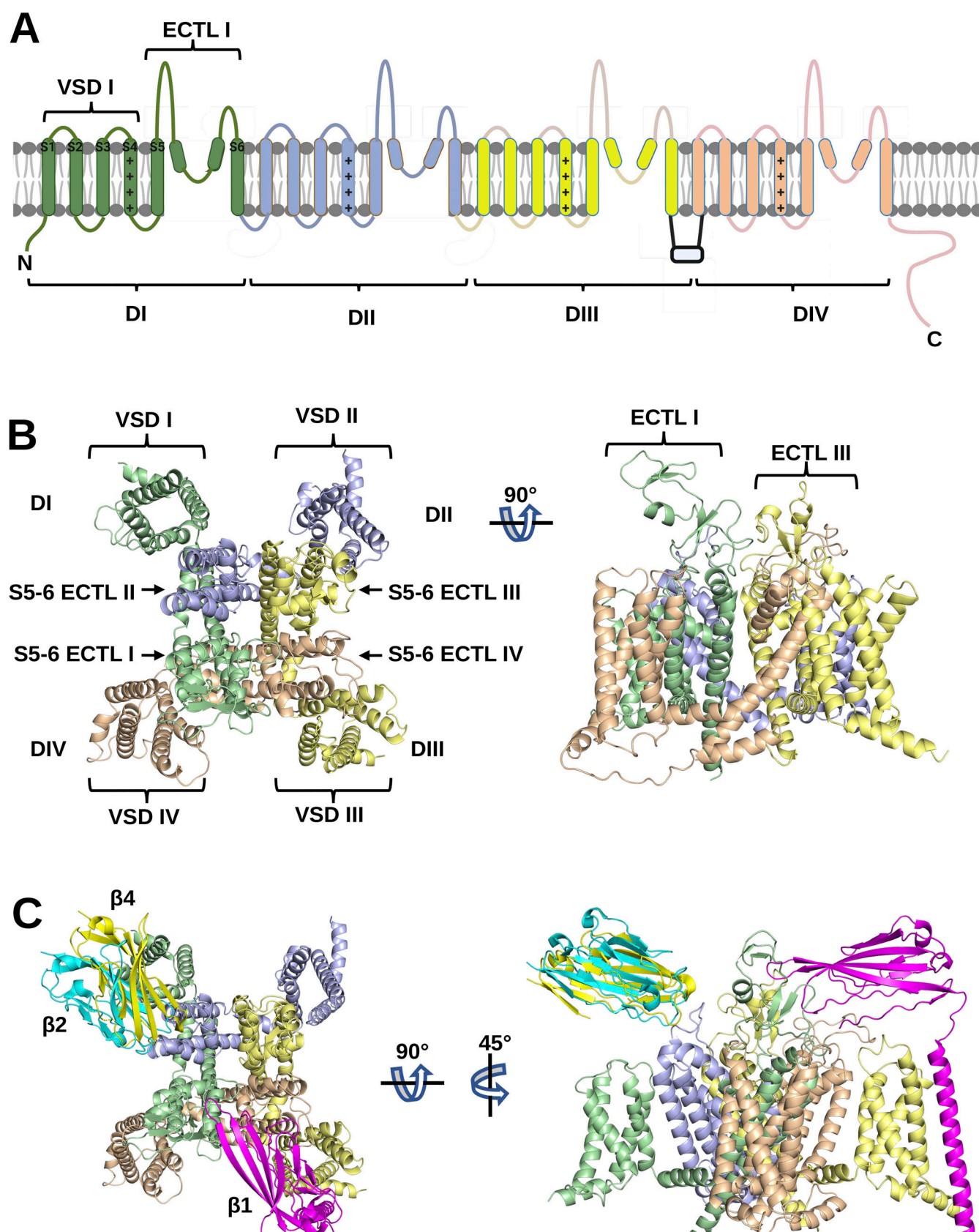

Figure 1.  **Na_V channel α-subunit domains and β-subunits. (A)** Key structural features of the Na_V channel α-subunit. The four internally homologous domains, DI–IV, are color-coded and labeled. Within DI, the transmembrane helices, S1–6 (including the positively-charged S4 helix), the voltage-sensing

domain (VSD), and the S5–6 extracellular turret loop (ECTL) are labeled. **(B)** Cryo-EM structure of human Na$_V$1.5 in top and side view (PDB: 7DTC). **(C)** The resolved binding sites of β1 (magenta), β2 (cyan), and β4 (yellow) based on cryo-EM structures of human Na$_V$1.1 (PDB: 7DTD); Na$_V$1.2 (PDB: 6J8E) and Na$_V$1.4 (PDB: 6AGF), as depicted from top and side views.

motifs are present in flexible regions of the ECTLs and, thus, were not resolved. Although absent from the resolved structures due to the flexibility of ECTL I, the primary sequence of the Na$_V$1.4 ECTL I shows the highest number of N-linked glycosylation motifs—seven—at this site. Among unique N-linked glycosylation motifs on the α-subunit surfaces, Na$_V$1.5 was found to have four total on domains I and III, Na$_V$1.6 has two on domain I, and both Na$_V$1.7 and Na$_V$1.8 have one on domains I and II, respectively. With the exception of the Na$_V$1.8 N819 residue on domain II, all N-linked glycosylation sites were noted on the ECTLs of domains I and III. The ECTL I is shown to have the highest density and largest variability of N-linked glycan motif start sites.

Structural modeling of the glycans may provide further insight into the positioning of the glycan trees relative to functional domains on the protein surface. Accordingly, representative glycan trees were modeled at each of the sites resolved on the Na$_V$1.1–1.8 cryo-EM structures. As shown in Fig. 3, A–G, the common sites found on domain I are located above the VSD of domain IV, and the common sites on domain III are found above the VSD of domain II around the pore. Interestingly, the glycan attached to N328 in Na$_V$1.5 and shared with all Na$_V$ channels, except Na$_V$1.7 and Na$_V$1.9, presents its negatively charged sialic acid moiety in proximity to the DIV VSDs (Fig. 3 C). Two of the three unique glycosylated residues on Na$_V$1.5 (N283, N318, N1388) are positioned above the VSD of domain III (Fig. 3 C). Interestingly, the increased presence of other glycans on domain I (N283, N288, N328) and domain III (N1365, N1374, N1380) may be sterically inducing a redirection of the three aforementioned glycans toward the area above the domain III VSD. The unique Na$_V$1.8 N-linked glycosylation site at N819 is positioned near the VSD of domain I—which is novel among the other Na$_V$ channels—and, thus, may interact with the VSD or nearby ECTLs (Fig. 3, E and G). The two conserved

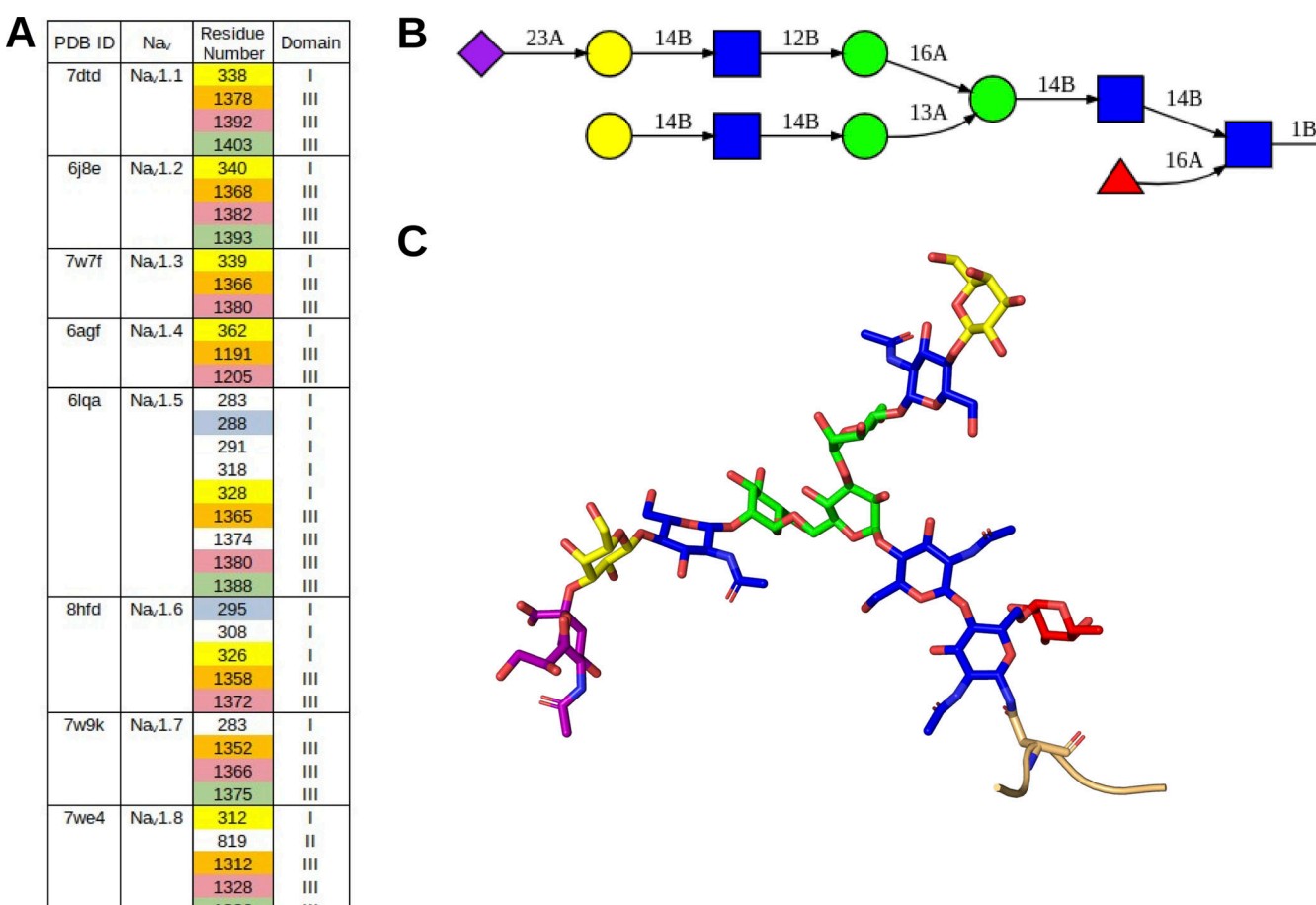

| PDB ID | Na$_V$ | Residue Number | Domain |
|---|---|---|---|
| 7dtd | Na$_V$1.1 | 338 | I |
| | | 1378 | III |
| | | 1392 | III |
| | | 1403 | III |
| 6j8e | Na$_V$1.2 | 340 | I |
| | | 1368 | III |
| | | 1382 | III |
| | | 1393 | III |
| 7w7f | Na$_V$1.3 | 339 | I |
| | | 1366 | III |
| | | 1380 | III |
| 6agf | Na$_V$1.4 | 362 | I |
| | | 1191 | III |
| | | 1205 | III |
| 6lqa | Na$_V$1.5 | 283 | I |
| | | 288 | I |
| | | 291 | I |
| | | 318 | I |
| | | 328 | I |
| | | 1365 | III |
| | | 1374 | III |
| | | 1380 | III |
| | | 1388 | III |
| 8hfd | Na$_V$1.6 | 295 | I |
| | | 308 | I |
| | | 326 | I |
| | | 1358 | III |
| | | 1372 | III |
| 7w9k | Na$_V$1.7 | 283 | I |
| | | 1352 | III |
| | | 1366 | III |
| | | 1375 | III |
| 7we4 | Na$_V$1.8 | 312 | I |
| | | 819 | II |
| | | 1312 | III |
| | | 1328 | III |
| | | 1336 | III |

Figure 2. **Position and structural representation of Na$_V$ glycans. (A)** The N-linked glycan positions on Na$_V$ channel cryo-EM structures are listed in a table format. Each color denotes homologous N-linked glycosylation sites within the indicated α-subunit isoform. **(B)** The representative N-linked glycan tree used in this study is shown in 2D format. **(C)** The corresponding 3D representation of the glycan tree is shown in B. N-acetylglucosamines are shown in blue, mannose in green, galactose in yellow, fucose in red, and sialic acid in pink. Connectivity (alpha = A, beta = B) between glycan moieties is notated (B).

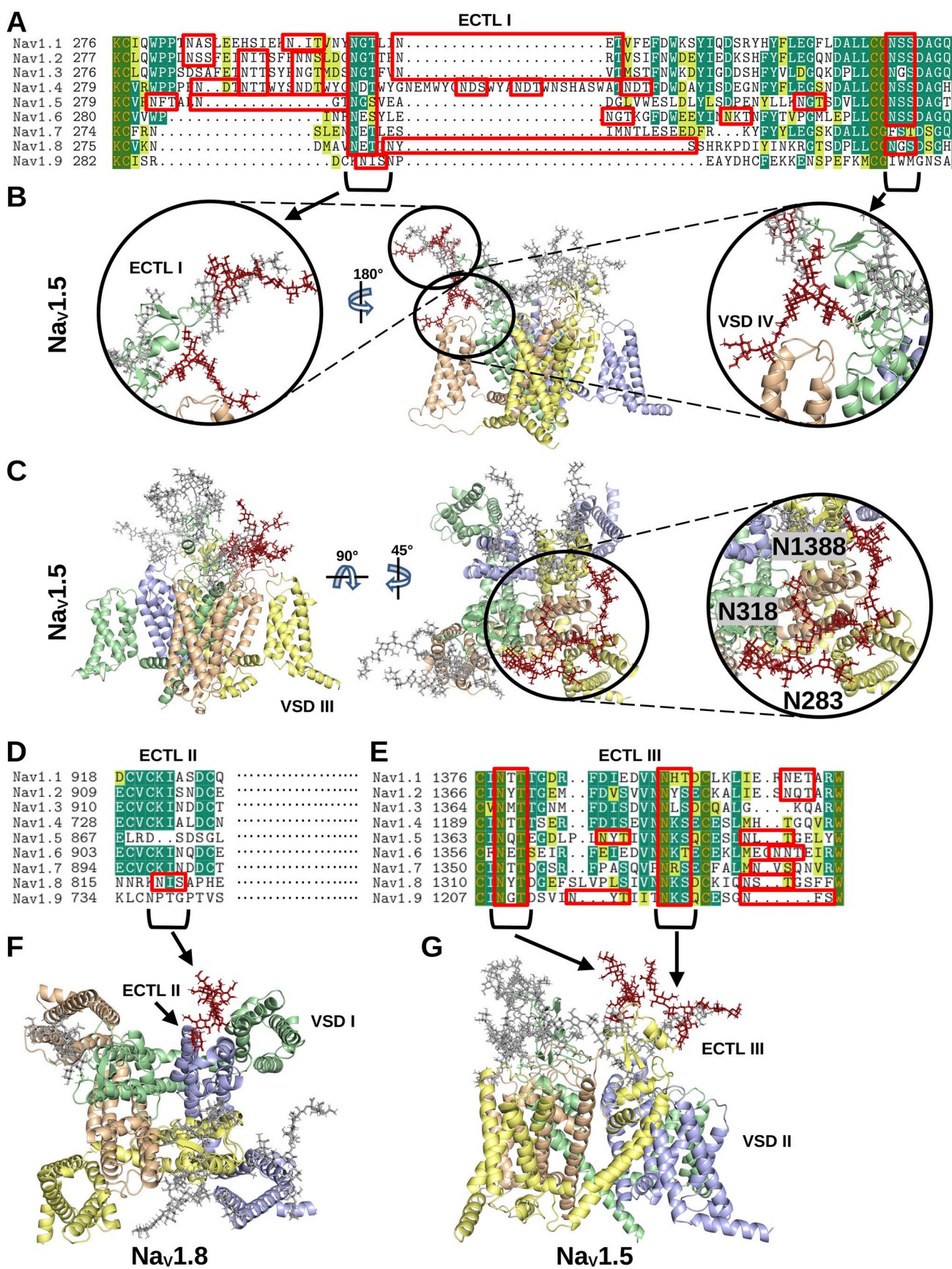

Figure 3. **N-linked glycans on Na$_V$ channels. (A)** Alignments of the amino acid sequences of the Na$_V$1.1–1.9 domain I ECTLs are shown. The residues are colored by similarity (yellow), conservation among 50% of sequences or higher (green), or 100% conservation (orange letters and in green blocks). All NX[S or

T] motifs in the sequence alignments are outlined with red rectangles to note the evolutionary conservation of the motifs. **(B and C)** The conserved glycans on the domain I ECTLs are colored crimson and shown on the modeled Na$_V$1.5 structure. In C, the unique Na$_V$1.5 glycans above VSD III are shown in crimson. **(D and E)** Alignments of the amino acid sequences of the Na$_V$1.1–1.9 domain II and III, respectively. Residue conservation and NX[S or T] motifs are colored as in A. **(F)** The unique Na$_V$1.8 N-linked glycan N819 is shown in crimson on the modeled Na$_V$1.8 structure. **(G)** The conserved glycans on domain III are colored in crimson and shown on the modelled Na$_V$1.5 structure.

glycans on ECTL III are shown to be near the pore (N1365 on Na$_V$1.5) and above VSD II (N1374 on Na$_V$1.5) (Fig. 3, F and H). Further structural dynamics may provide additional insight into the flexibility and, thus, density of the glycans on the Na$_V$ channel surface.

### Structural effect of Na$_V$ channel N-linked glycans on β-subunit accessibility

Na$_V$ channels are involved in regulating the action potential of diverse cell types in the nervous and muscular systems. For example, Na$_V$1.4 and Na$_V$1.5 are known for propagating action potentials in skeletal and cardiac muscle, respectively (Loussouarn et al., 2016). Although differences in protein sequence and structure between Na$_V$1.4 and Na$_V$1.5 have been investigated to better understand the physiological differences between muscle types (Salvage et al., 2023a), comparing the differences in glycan conformational landscape may shed light on the functional differences that permit different types of action potential propagation specific to the tissue. Furthermore, Na$_V$1.4 shares similar glycosylation profiles with Na$_V$1.1, Na$_V$1.2, and Na$_V$1.3, and thus, may serve as a representative of such structures. On the other hand, Na$_V$1.5 has a much higher density of glycans on its surface and further analysis may provide insights into the conformational landscape of such extensive glycosylation. Therefore, to investigate glycan tree flexibility and interactions, both at sites shared between isoforms and for Na$_V$1.5-specific sites, all-atom molecular dynamics simulations were conducted with the glycosylated and non-glycosylated Na$_V$1.4 and Na$_V$1.5 channels in a representative plasma membrane (Fig. 4, A–F).

The molecular dynamics simulations reveal a broad landscape of conformations and interactions displayed by the glycans. Importantly, in both channels, the pore is not blocked by glycans. The lack of density above the channel opening suggests that ion conductance would not be hindered by the presence of glycosylation sites near the pore (e.g., N1191 on Na$_V$1.4 and N1365 on Na$_V$1.5). In contrast, for both channels, the ECTLs of domains I and III are completely covered by glycans on their corresponding loops and the ECTL of domain II is largely overshadowed by the glycans on the ECTL of domain I. The VSDs of domains II and IV were discovered to potentially interact with glycans, which occurred in both Na$_V$1.4 and Na$_V$1.5 α-subunits (Fig. 4, B and E). Notably, in the simulations, the mannose, galactose, and N-acetylglucosamine moieties more than the sialic acid tips remained closer to the basic residues within the VSD throughout the simulations. Based on the simulation data, which shows that the sialic acids interacted with channel residues and membrane lipids surrounding the inner portion of the VSD, the sialic acids may, nevertheless, interact with residues within the VSD and affect gating and kinetics of the channel. The glycans

pointing parallel to the membrane from the ECTL of domain III (i.e., N1205 on Na$_V$1.4 and N1380 on Na$_V$1.5) were not found to interact with the VSD of domain II and were more likely found to interact with membrane head groups in the space between the VSDs of domains II and I (Fig. 4, C and F).

The conformations of the glycan trees over the course of the simulations were also compared with known β-subunit binding sites to determine how their positions may affect binding. Here, analysis of the total 3D space occupied by the glycan conformations revealed a notable difference between Na$_V$1.4 and Na$_V$1.5, as depicted in Fig. 4, A and D, respectively. Specifically, the ECTL of domain IV and the VSM of domain III are exposed in Na$_V$1.4 but are completely covered in Na$_V$1.5. This isoform-specific difference reflects the presence of Na$_V$1.5-unique glycosylation sites at N283 and N318, together with another glycan, N1388, on the domain III ECTL, which primarily points toward the ECTL of domain IV. These domains III and IV regions correspond to the binding site of the β1 Ig-domain in Na$_V$1.4 (Fig. 5, A and C) (Pan et al., 2018). Currently, there are no cryo-EM structures for Na$_V$1.5 and β-subunits (Li et al., 2021; Jiang et al., 2020). But, mutagenesis evidence suggests that the β1 subunit transmembrane domain binds close to Na$_V$1.5 domain III VSD (Zhu et al., 2017), possibly in a manner similar to its binding in Na$_V$1.3 (PDB: 7w77), Na$_V$1.4 (PDB: 6agf), Na$_V$1.6 (PDB: 8GZ1), and Na$_V$1.7 (PDB: 7w9k). However, unlike these other Na$_V$ isoforms, the presence of Na$_V$1.5 glycans at N283, N318, and N1388 will cover the β1 Ig-domain binding (Fig. 5, D and F). Hence, the β1 subunit will still attach to Na$_V$1.5 via its transmembrane domain, but it will not be possible for the Ig-domain to bind Na$_V$1.5 in the same way it binds to Na$_V$1.4 (see next section, below).

The VSD of domain I and ECTL of domain II have been shown to be the binding site of β2 and β4 Ig-domains (Salvage et al., 2020a). As shown in Fig. 5, A, B, D, and E, in both Na$_V$1.4 and Na$_V$1.5, the glycans largely do not cover the space at these sites. Thus, glycans will not prevent the binding of the β2 and β4 Ig-domains to Na$_V$1.5. Note, by contrast, that Na$_V$1.8 possesses a unique glycan at N819, which reveals a direct clashing interaction with the putative binding sites of β2 and β4 Ig-domains (Fig. 3 F).

### Na$_V$1.5 N-linked glycans may facilitate homophilic interactions between β-subunits

Previous reports have indicated that β1-subunits can interact to permit trans interactions between cells (Malhotra et al., 2000). Our structural analysis of N-linked glycans near the Na$_V$1.5 β1 binding site indicates that the β1 Ig-domain will be unable to interact with the channel. Considering that the β-subunit Ig- and transmembrane domains are connected via a flexible linker, we propose that in Na$_V$1.5 channels, the β1 Ig-domain is less

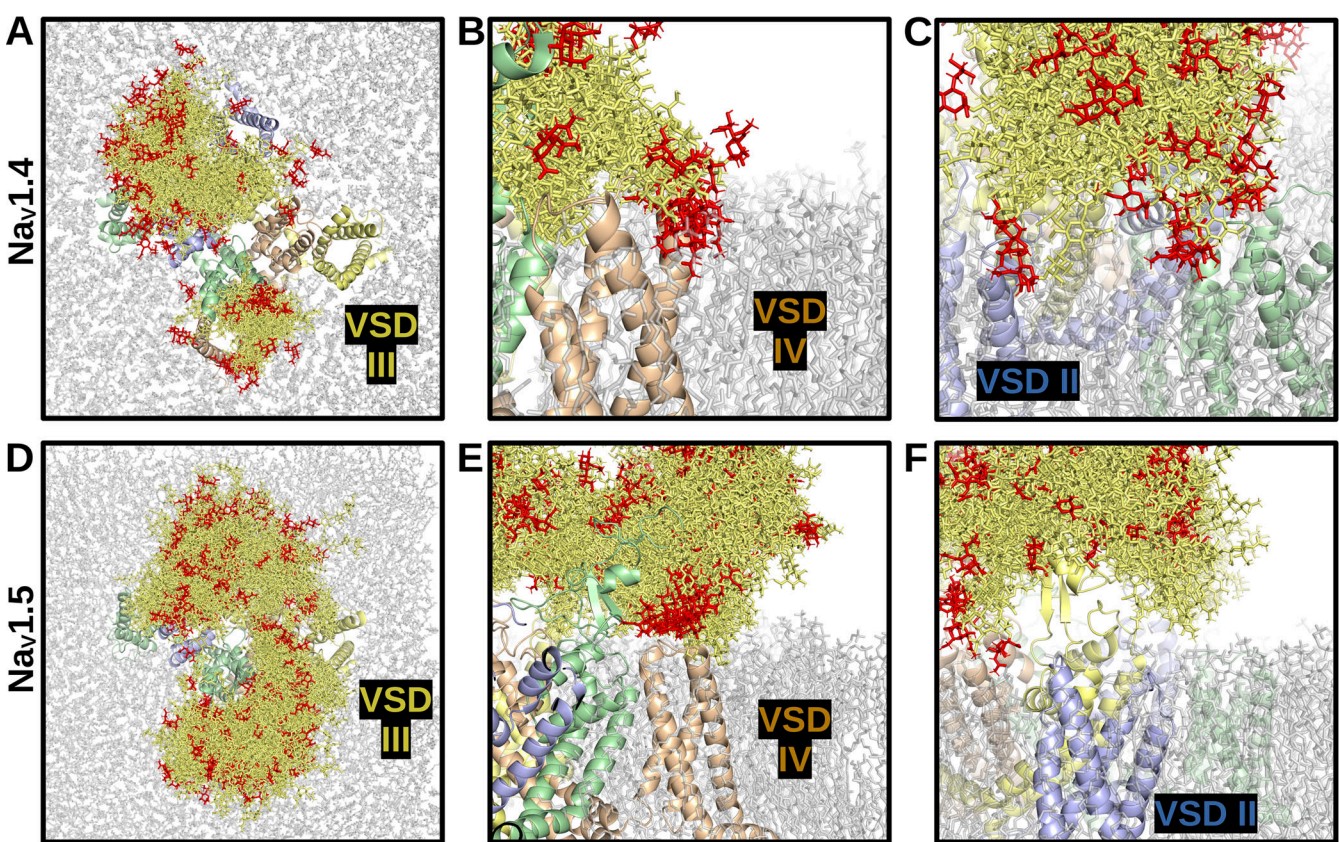

**Figure 4.    N-linked glycan density on Na_V1.4 and Na_V1.5.** Images show overlapping and superimposed snapshots of glycan conformations extracted from the molecular dynamics simulations every 5 ns, over a time frame of 150 ns. **(A)** Top view of Na_V1.4. **(B and C)** Side views of Na_V1.4, VSDs IV and II, respectively. **(D)** Top view of Na_V1.5. **(E and F)** Side views of Na_V1.5, VSDs IV and II, respectively. Glycans interacting with VSD IV are depicted. The Na_V1.4 (C) and Na_V1.5 (F) glycans interacting near VSD II are displayed. Glycans are shown in dark gold, with sialylated tips in red.

structurally constrained and will be redirected outwards from the channel, which may facilitate the formation of homophilic trans interactions between adjacent membranes expressing Na_V1.5/β1.

One case in particular where Na_V1.5 and β1 might be engaging in trans homophilic interactions is in the perinexal space of cardiomyocytes (Veeraraghavan et al., 2018). The cardiomyocyte perinexus is a 100–200 nm-long and 15–30 nm-wide anatomically distinct region within intercalated discs that connects cardiomyocytes and is bounded by connexin-containing gap junctions (Hoagland et al., 2019). Na_V1.5 α-subunits associated with the β1-subunit have been shown to cluster on opposing membranes, particularly close to the connexins. It has been suggested that this clustering is facilitated by trans interactions mediated by homophilic binding between mutually opposed β1-subunit Ig-domains (Salvage et al., 2023b). The potential structural assembly of such a complex has not yet been explored. Therefore, using protein structure alignment, modeling, and docking methodologies, the structural implications of a Na_V1.5/β1 complex interacting in trans with another Na_V1.5/β1 complex were explored.

To investigate the structural assemblies that would permit such potential trans interactions, two β1 Ig-domains were, first, docked to one another to find the most likely oligomeric states. Two clusters of dimeric β1 interactions were consistently predicted using different tools: (1) tip-to-tip, wherein the turn

regions distal from the transmembrane helix, which are primarily involved in complexing with the Na_V channels, are interacting (Fig. 6, A and B), and (2) side-to-side, in which the β-strands interact (Fig. 6, C and D). The predicted binding affinity of the dimer interactions, using PRODIGY (Xue et al., 2016), was –8.1 kcal/mol for the tip-to-tip model and –10.1 kcal/mol for the side-to-side model, suggesting that side-to-side interactions may interact more strongly. The two β1 Ig-domains of both the tip-to-tip and side-to-side models were then pulled apart from the most C-terminal residue, M154, of each flexible linker with steered (constant velocity pulling) molecular dynamics simulations to determine the maximum perinexal intermembrane distance based on theoretical biophysical constraints. The β1 dimer structures that were selected to represent the maximum distance were extracted 10 ps before dimer interface rupture. Upon measuring the predicted distance between the lipid head groups, the tip-to-tip model shows a distance of 18.72 nm and the side-to-side model shows 15.34 nm. Both predicted intermembrane distances agree with the experimentally derived 15–30 nm between membranes that form the cardiomyocyte perinexal space (Rhett et al., 2012; Veeraraghavan et al., 2018). These data corroborate with previously published data that suggest that trans interactions between β1 Ig-domains may be responsible for establishing the structural restraints of the cardiomyocyte perinexus (Gourdie, 2019).

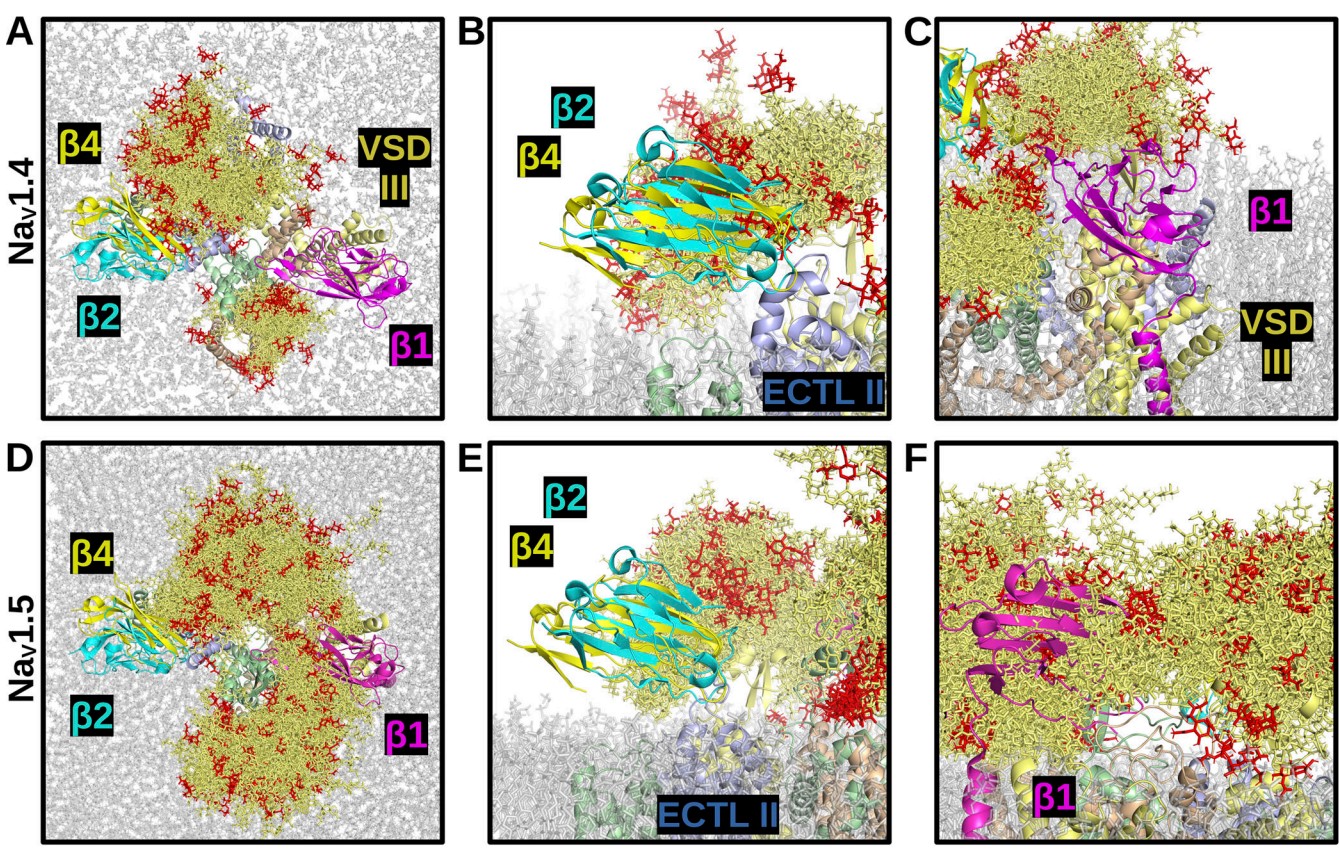

**Figure 5.** **Effect of N-linked glycan density on β-subunit binding for Na_V1.4 and Na_V1.5.** Images show overlapping and superimposed snapshots of glycan conformations from the molecular dynamics simulations, extracted every 5 ns, over a time frame of 150 ns. **(A)** Top view of Na_V1.4, with β1 (magenta), β2 (cyan), and β4 (yellow). **(B)** Side view of Na_V1.4 interacting with β2 and β4. **(C)** Side view of Na_V1.4 interacting with β1. **(D)** Top view of Na_V1.5, with β1, β2, and β4. **(E)** Side view of Na_V1.5 interacting with β2, and β4, assuming the same binding as with Na_V1.4. **(F)** Side view of Na_V1.5 interacting with β1, assuming the same binding as with Na_V1.4. Glycans are shown in dark gold, with sialylated tips in red.

Na_V β3 has been shown to trimerize and potentially permit homophilic cis interactions (Namadurai et al., 2014). Since the β3 Ig-domain would be redirected in the same manner as β1, both β-subunits may bind at VSD III through interactions between transmembrane domains and create a network of Na_V channel supra-clustering on both membranes in the perinexal space. If the transmembrane domains of β2 and β4 do, indeed, interact stably with the Na_V1.8 transmembrane domains, cis homophilic dimerization—in which β-strands in the β4 Ig-domains exchange with one another to covalently link the dimer—between β4 subunits may also be relevant in the case of N-linked glycosylation of the ECTL on domain II of Na_V1.8. Such dimerization may link Na_V1.8 channels together to form higher-order structures in the peripheral nervous system. Therefore, N-linked glycans on Na_V α-subunits may induce higher-order assembly of Na_V channels in specialized membranes that contribute to physiological mechanisms, such as ephaptic conduction (Hoagland et al., 2019).

### Clinically relevant mutations in Na_V channel N-linked glycosylation motifs

Further evidence for the functional roles of glycan trees on the Na_V channel structures may be revealed by the presence of mutations that both disrupt N-linked glycosylation and manifest into clinical pathologies. Mutations that remove glycosylation sites may occur at the glycosylated asparagine itself, the serine/threonine at the N+2 position, or at the N+1 position if a proline is substituted for the X residue. Thus, disease-causing mutations from the "Disease/Phenotypes and variants section" of each Na_V channel UniProt page, an extensive review of Na_V variants compiled by Huang et al. (2017b), and ClinVar (Landrum et al., 2020) were referenced to notate mutations at N-linked glycosylation sites are associated with pathological phenotypes. Only ClinVar entries classified as "Likely Pathogenic" and "Pathogenic" were considered. The mutations are summarized in Fig. 7 A.

Na_V1.1 has been discovered to be distributed throughout the central nervous system (Duflocq et al., 2008) and peripheral nervous system (Espino et al., 2022). All four N-linked glycosylation sites—N338, N1378, and N1392—on Na_V1.1 were affected by seven mutations—S340F (Depienne et al., 2009), S340P (Depienne et al., 2009), S340Y (Landrum et al., 2020), N1378H (Zuberi et al., 2011), N1378T (Zuberi et al., 2011), H1393P (Stefanaki et al., 2006), T1394I (Zuberi et al., 2011)—in Dravet syndrome patients. Three clinically relevant frameshift mutations—N1392fs, H1393fs, and T1394fs—occurred at the N1378 site and were associated with Dravet syndrome, but the recorded clinical outcomes likely stem from downstream effects

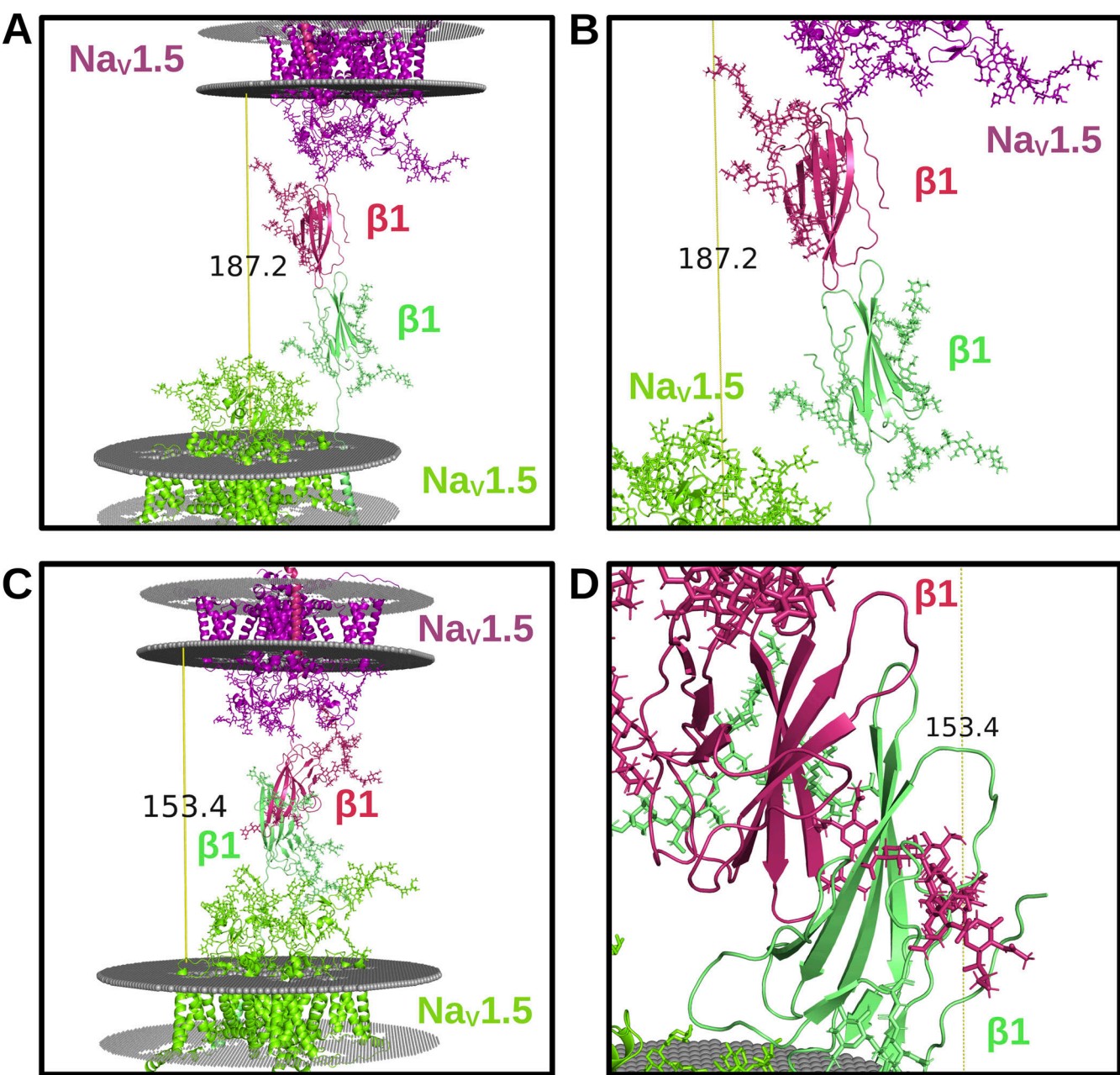

Figure 6. **Putative trans interactions between Na$_V$1.5/β1 complexes. (A and B)** Full snapshots (A) and close-up (B) image of the "tip-to-tip" models of the trans interactions between Na$_V$1.5/β1 complexes on opposing *membranes* (grey). **(C and D)** Full snapshots and (D) close-up image of the "side-to-side" models. Note the potential for glycan hindrance in the side-to-side model.

of the frameshift mutations on protein structure that are unrelated to glycosylation. The mutations at S340 and H1393/T1394 may prevent interactions between the glycan and the VSDs of domains IV and II, respectively, which brings further attention to the putative interactions between the negatively charged glycan sialic acids and the positively charged VSD residues. The N1378H and N1378T variants may disrupt the incorporation of the glycan located above the pore, which may affect several processes, such as gating, trafficking, or protease inhibition. These mutations (Fig. 7 B) affect all but one glycosylation site on Na$_V$1.1, which reveals the selective functional roles for glycosylation in regulating neural Na$_V$ channel activity.

Na$_V$1.5 is associated with cardiomyocyte action potential propagation (Rook et al., 2012). Two sites—N318 and N1380—were affected by two mutations—T320N (Kapplinger et al., 2010) and S1382I (Kapplinger et al., 2010; Smits et al., 2002)—on Na$_V$1.5 in Brugada syndrome patients (Fig. 7 C). The T320N mutation removes the glycan that is centrally positioned among the other glycans for preventing interactions with the β1 and potentially β3 Ig-domains. These data potentially provide further evidence that blocking the β-subunit Ig-domains from binding to the channel is vital for cardiomyocyte function. The Na$_V$1.5 S1382I mutation may prevent interactions from the glycan that is located above the VSD of domain II. These data

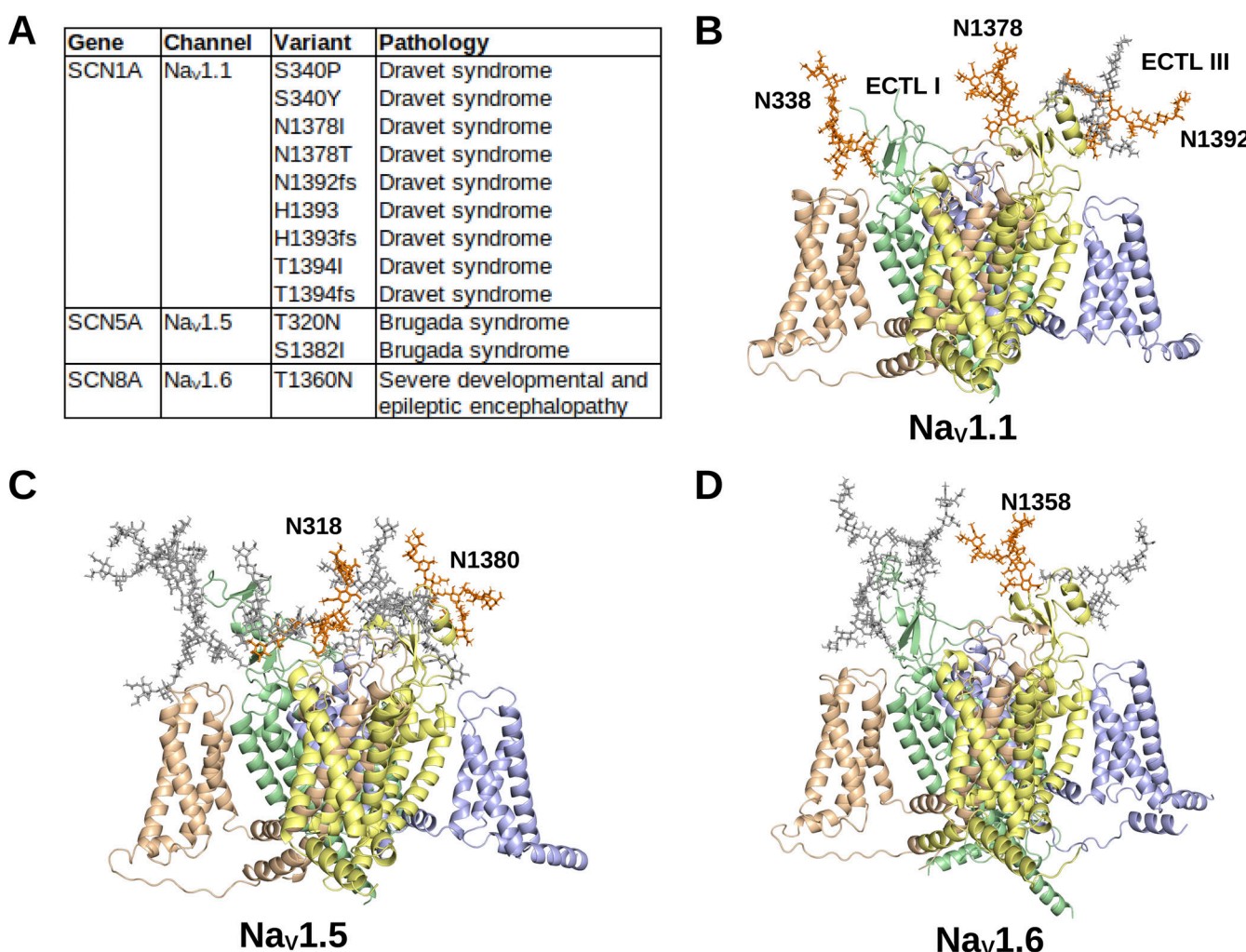

**Figure 7.  Mutations at N-linked glycan sites linked to clinical pathologies. (A)** The N-linked glycan sites disrupted by mutations and their associated pathologies are listed. **(B–D)** The N-linked glycans affected by mutations (orange) in (B) Na$_V$1.1; (C) Na$_V$1.5 and (D) Na$_V$1.6 are shown amongst the other N-linked glycans (grey) on the Na$_V$ channel surfaces.

further add to the notion that glycans may interact and modulate the structure–function relationship of VSDs.

Na$_V$1.6 is widely expressed throughout the brain and regulates the initiation of action potentials at nodes of Ranvier (Wagnon et al., 2016). Only one mutation—T1360N—was found to interfere with N-linked glycosylation and was associated with global developmental delay and seizures (Fig. 7 D) (Wengert et al., 2019). Similar to the mutations at N1378 in Na$_V$1.1, the Na$_V$1.6 T1360N variant may prevent glycosylation above the pore, which could affect ion conduction and gating.

The presence of variants associated with human pathologies at N-linked glycosylation sites provides additional support for the critical roles of such glycans in the structure and function of Na$_V$ channels. Mutations at specific N-linked glycans may also shed light on the importance of the position of the glycans on the Na$_V$ channel surfaces. Among the three sites conserved by all Na$_V$ channels, mutations in all three were independently associated with a clinical pathology in Na$_V$1.1, and only one conserved site was affected by a mutation in pathologies related to Na$_V$1.5 (T320) and Na$_V$1.6 (T1360). The other recorded clinically

relevant Na$_V$1.5 mutation is unique to Na$_V$1.5 and, as reported above, may affect interactions with the β1 and β3 subunit Ig-domains. Of note, synonymous mutations at N-linked glycosylation sites were reported in Na$_V$1.3, Na$_V$1.5, Na$_V$1.6, Na$_V$1.7, and Na$_V$1.8 in the ClinVar data, all of which were classified as "Benign" or "Likely Benign." Further investigation into the clinical relevance of Na$_V$ channel N-linked glycans may provide mechanistic insights into their functional roles, and biophysical experimentation of the effect of N-linked glycans on Na$_V$ channels may guide clinical and give rise to novel diagnostic and therapeutic options.

## Discussion

Glycosylation of Na$_V$ channels has been implicated in altering biophysical properties, such as gating, trafficking, and localization (Zhang et al., 1999; Cortada et al., 2019; Ednie and Bennett, 2012; Wang et al., 2021). Although several studies have outlined the effects of differing composition and amounts of glycosylation in vitro and in vivo, little work has been conducted regarding the

structural effect of N-linked glycans on Na$_V$ channels and their impact on interactions between the channels and accessory β-subunits. Therefore, herein, the potential interactions, conformational landscape, and clinical effects of N-linked glycans on Na$_V$ channels extracellular surfaces were investigated using sequence and structure-based bioinformatics methodologies.

Sequence and structure comparisons revealed three glycan sites that were found to be common to all Na$_V$ channels, which are located on the ECTL of domains I and III. Interestingly, two of the three sites are positioned directly above the VSDs of domains II and IV. Glycan sialic acid residues are known to influence the electric field sensed by the VSDs (Ednie and Bennett, 2012). Since negatively charged sialic acid moieties have been documented to be present on the termini of glycan tree branches, the voltage-sensing properties of the channels may be altered due to interactions between the positive residues in the VSD and glycans (Robinson et al., 2023). The remaining glycan is found nearly above the channel pore on the ECTL of domain III. However, since the glycan is directed outwards from the channel, it may function to modulate or prevent interactions with other proteins. Interestingly, in examining the sites on cryo-EM structure of the electric eel Na$_V$1.4, which is the only other available resolved non-mammalian vertebrate Na$_V$ channel structure, all three conserved human N-linked glycosylation sites are the only sites present (Yan et al., 2017). These data point to deeply evolutionarily conserved functional roles for glycosylation at these three sites.

Na$_V$1.5 was recorded to contain four unique motifs, which were widely distributed on the ECTLs of domains I and III. Notably, two glycans (N283 and N318) on domain I and one glycan (N1388) on domain III were found to largely occupy the area above the VSD of domain III, which, thus, directly covers the binding site of the β1 Ig-domain. The unique glycans on domains I and III of Na$_V$1.5 may, thus, prevent the β1 Ig-domain from interacting with the channel. Although the Ig-domain is itself rigid, it is connected to the transmembrane helix via a flexible, disordered neck (Salvage et al., 2020a). Hence, a β1-subunit bound to a fully glycosylated Na$_V$1.5 α-subunit via its transmembrane helix will impart a greater degree of conformational flexibility on the Ig-domain than would otherwise be the case. One example where this may be physiologically important is within the opposing perinexal membranes of the intercalated discs between adjacent cardiomyocytes, where Na$_V$1.5 and β1-subunits are tightly packed (Veeraraghavan et al., 2018). The perinexal space between these membranes has been shown to facilitate ephaptic conduction between cardiomyocytes (Hichri et al., 2018). Mathematical modeling studies have revealed that the distance between membranes is critically important for ephaptic conduction, no more than about 20 nm is optimal (Lin and Keener, 2010; Mori et al., 2008). The β1-subunit Ig-domain is known to be capable of trans-mediated cell adhesion, and inhibition of β1-mediated trans adhesion within the perinexus leads to reduced ephaptic conduction (Veeraraghavan et al., 2018). The distance between the membranes that define the perinexal space is about 15–20 nm (Veeraraghavan et al., 2018). But, in many places, particularly where closer to the connexin gap junctions, this distance is less

(2–10 nm). Our modeling identified two possible modes of trans homophilic binding between β1 Ig-domains: "tip-to-tip" or "side-to-side" with estimated membrane-to-membrane distances of 18.72 and 15.34 nm, respectively (Fig. 6). But, these are maximum estimates, assuming the flexible neck connecting the Ig- and transmembrane domains is fully extended. Thus, both models are consistent with the morphological data. An inhibitory peptide was described by Veeraraghavan et al., which successfully disrupted the trans-homophilic binding between β1 Ig-domains and would be expected to interfere with a side-to-side-type interaction between Ig-domains (Veeraraghavan et al., 2018). However, the β1 Ig-domain contains a glycan at N135 and in the side-to-side model, which points directly into the other β1 Ig-domain (Fig. 6). This would make a side-to-side model of the type identified here highly unlikely. Thus, the tip-to-tip model may be more likely to represent a stable state for trans interactions. As further support, a tip-to-tip model has also been proposed for cis homophilic interactions between β3 subunits (Glass et al., 2020).

The analysis also identified an N-linked glycan unique to Na$_V$1.8 on the ECTL of DII. Notably, the unique glycan at this site would extend over the area above the VSD of DI, thus occluding the binding sites of the β2 and β4 Ig-domains. Therefore, such glycosylation may either prevent interactions with the β-subunits entirely—since the transmembrane interactions are likely to be more transient than β1 as noted by their absence in resolved cryo-EM structures—or the Ig-domains may be redirected as in the proposed case of β1 and potentially β3. It should further be noted that Na$_V$1.8 and Na$_V$1.5 both lack a free cysteine residue on domain II ECTL and thus cannot form the covalent disulfide bond with the β2 and β4 Ig-domains that occurs with most Na$_V$ channels (Salvage et al., 2020a). Interestingly, the β4 Ig-domain can form cis-interacting covalent dimers using the cysteine residue that forms a disulfide bond to most Na$_V$ α-subunits (Shimizu et al., 2017). Hence, the absence of a suitable cysteine residue on Na$_V$1.5 and Na$_V$1.8, together with a blocking glycan on Na$_V$1.8, would further facilitate cis-interactions between β2 and β4 (Salvage et al., 2020a).

The ECTL I of Na$_V$1.4 was found to be the most densely glycosylated region with seven N-linked glycan sites among all human Na$_V$ channels. Previous studies demonstrated that sialylated N-linked glycans on the Na$_V$1.4 ECTL I uniquely affect channel gating (Bennett et al., 1997; Bennett, 2002; Ednie et al., 2015). Notably, patch-clamp electrophysiology of a chimera of Na$_V$1.5 with the ECTL I of Na$_V$1.4 resulted in the shifting of all voltage-dependent parameters with respect to wild-type Na$_V$1.5, while replacing the ECTL I of Na$_V$1.4 with that of Na$_V$1.5 resulted in no gating changes under reduced sialylation (Bennett, 2002). Furthermore, changes in the expression of sialyltransferases (both artificially and during development) and enzymatic cleavage of sialic acids have been shown to affect Na$_V$ channel gating (Stocker and Bennett, 2006; Ednie et al., 2013). These data highlight the significant effect of position, location, and extent of sialylation of N-linked glycans on Na$_V$ channel surfaces. Considering that Na$_V$1.4 is specifically expressed in skeletal muscle cells, the packing of negatively charged sialic acids onto one extracellular loop may lead to unique tissue-specific effects on

action potential propagation. Also, the glycan cloud on the ECTL I extends near but not over the channel pore; therefore, sialic acids on the tips of the glycan trees may affect sodium ion conductance through the channel by contributing to the negative membrane potential or interacting with sodium ions directly (Bennett et al., 1997). Future work looking into the biophysical effects of these uniquely nested N-linked glycans on Na$_V$1.4 may shed light on the roles of negatively charged glycans on Na$_V$ and other voltage-gated ion channels.

The β-subunits affect the electrophysiological properties of the Na$_V$ channel α-subunits (Namadurai et al., 2015). Binding of the β-subunits modulates key Na$_V$ channel conformational dynamics during an action potential cycle, including shifting the voltage ranges of activation and inactivation and altering the rate of recovery from inactivation (Yu et al., 2005; Cusdin et al., 2010; Cummins et al., 2001). These effects vary with different α- and β-subunit combinations. The Ig-domains of the β-subunits also contain N-linked glycosylation sites. Notably, sialic acids on the Ig domains have been found to shift Na$_V$ channel voltage-gating parameters (Johnson et al., 2004). Intra- and intersubunit interactions between α- and β-subunit glycans may, thus, influence either the structural conformational dynamics or further contribute to changes in membrane potential via the additional negatively charged sialic acid moieties on the β-subunit N-glycans. Hence, these data may have important implications for understanding Na$_V$1.5 and Na$_V$1.8 channel gating (Vijayaragavan et al., 2004; O'Malley and Isom, 2015).

Distinct glycosylation states of Na$_V$ channel α-subunit isoforms, including Na$_V$1.5, have been detected (Mercier et al., 2015; Laedermann et al., 2013). N-linked glycosylation and Na$_V$ channel oligomerization occur contemporaneously within the endoplasmic reticulum at either independent or co-dependent rates (Braakman and Hebert, 2013). Glycosylation of Na$_V$1.5 α-subunit at N318 and N283 will prevent the binding of the β1 Ig-domain to DIII VSD, but not its association with the α-subunit via its transmembrane helix. But if the Na$_V$1.5 α-subunit assembles with β1 before it is glycosylated, would that permit the binding of the β1 Ig-domain to the DIII VSD? If so, then a cell co-expressing Na$_V$1.5 α- and β1-subunits may contain a mixed population of Na$_V$1.5/β1 hetero-oligomers with distinct structural properties and perhaps distinct functional behavior (Schoberer et al., 2018; Ninagawa et al., 2021). Their proportion would depend on the relative rates of glycosylation versus hetero-oligomer assembly. The Na$_V$ channel glycosylation patterns and rates may also be dependent on the expression of specific glycosyltransferases and, thus, may be cell or tissue-specific (Medzihradszky et al., 2015). Further work is required to underpin the competition dynamics between β-subunit binding and glycosylation of Na$_V$ channel α-subunits within the endoplasmic reticulum.

The conformational variability of the Na$_V$ channel glycans creates a notable shield around the Na$_V$ channel surfaces (Seitz et al., 2020). In addition to altering the binding sites of β-subunits, glycans on Na$_V$ channel surfaces may interfere with binding to toxins, proteases, and other proteins (Beaudoin et al., 2022). Additionally, consideration of the glycan structural conformational space may guide studies targetting Na$_V$ channels with inhibitory or activating small molecules, peptides, and antibodies (Seitz et al., 2020). For example, the spider toxin Dc1a binds to the DII VSD of Na$_V$1.7, which is in agreement with the data described above considering that the DII VSD is largely unoccupied by glycans (Bende et al., 2014). Furthermore, incorporating isoform-specific glycosylation structural information may further inform the targeting of specific Na$_V$ channels. For example, the glycan overshadowing the DIV VSD of Na$_V$1.5 (N328) is conserved among Na$_V$ channels with the exception of Na$_V$1.7 and Na$_V$1.9. Indeed, the DIV VSD of Na$_V$1.7 has been shown to confer isoform-specific targeting of Na$_V$1.7 and is also the binding site of venom toxins (e.g., OD1) specific to Na$_V$1.7 (Kschonsak et al., 2023; Salvage et al., 2023c; Jalali et al., 2005). Therefore, alongside the electrostatic and structural effects, further investigation into the roles that glycans play in modulating interactions between Na$_V$ channels and other binding partners may reveal new insights into higher-order complexes at the cell surface, coordination of intracellular signaling pathways, and site-specific drug target selection.

In conclusion, herein, molecular modeling and all-atom molecular dynamics simulations were applied using the resolved Na$_V$ channel α-subunit structures and the resolved sugar moieties as references. In particular, a comparative analysis of the skeletal muscle-specific channel Na$_V$1.4 and the heart muscle-specific channel Na$_V$1.5 cryo-EM structure sites was conducted to better understand the landscape of glycan conformations with respect to Na$_V$ channel domains and β-subunit binding. Molecular dynamics simulations revealed that negatively charged sialic acid residues of two conserved glycans may interact with the VSDs of domains IV and II. Notably, three of the five Na$_V$1.5 isoform-specific N-linked glycosylation sites cover the landscape above domain III where the β1 (and likely β3) Ig-domains bind in other Na$_V$ channel isoforms. These glycans will thus prevent the binding of the β1 and β3 Ig-domains, allowing them to be redirected outwards and to rotate more freely while preserving transmembrane domain interactions with the domain III VSD. It was also noted that Na$_V$1.8 contains a unique N-linked glycosylation site on domain II ECTL that likely prevents interactions with the Ig-domains of β2 and β4. Previously determined complexing between the β1 transmembrane domain and Na$_V$ channel VSD III transmembrane domains suggests that β1 and β3 may, nevertheless, bind to Na$_V$1.5 although the Ig-domain would be directed outwards from the channel. Protein–protein docking revealed that this blocked interaction may, thus, redirect the Ig-domain outward for more likely interactions with other β3 Ig-domains to permit cis supraclustering of channels or the β1 Ig-domains for trans interactions between channels on opposing plasma membranes, such as those in the cardiomyocyte perinexal space. Further experimental work is necessary to validate these hypotheses. We propose that the isoform-specific structural features of Na$_V$1.5 and Na$_V$1.8 may have evolved to facilitate functional interactions, which would include the promotion of β-subunit-induced trans and cis crosslinking.

## Materials and methods

### Sequence accessions and analysis

The amino acid sequences of Na$_V$1.1 (UniProt accession: P35498), Na$_V$1.2 (Q99250), Na$_V$1.3 (Q9NY46), Na$_V$1.4 (P35499), Na$_V$1.5 (Q14524), Na$_V$1.6 (Q9UQD0), Na$_V$1.7 (Q15858), Na$_V$1.8 (Q9Y5Y9), and Na$_V$1.9 (Q9UI33) were retrieved from UniProt (UniProt Consortium, 2023). Multiple sequence alignments were constructed using the ClustalW algorithm (Thompson et al., 1994) implemented in the R (version 4.1.2) package "msa" (Bodenhofer et al., 2015).

### Protein structure, glycan, and membrane modeling

The experimentally resolved structures of Na$_V$1.1–Na$_V$1.8 were downloaded from RSCB PDB (Burley et al., 2023): Na$_V$1.1 (PDB ID: 7DTD), Na$_V$1.2 (6J8E), Na$_V$1.3 (7W7F), Na$_V$1.4 (6AGF), Na$_V$1.5 (6QLA), Na$_V$1.6 (8FHD), Na$_V$1.7 (7W9K), and Na$_V$1.8 (7WFW). The NX[S or T] motifs resolved with at least one glycan moiety were considered for glycan modeling. Protein structures were visualized using PyMol (Schrödinger, LLC and DeLano, 2020).

A representative N-linked glycan tree structure (supporting data under peer review) was modeled onto each NX[S or T] motif using the CHARMM-GUI Glycan Reader and Modeller (Park et al., 2019). The glycosylated models of Na$_V$1.4 and Na$_V$1.5 were placed in a representative mammalian cell membrane, as outlined by Ingólfsson et al. (2014, 2017), using PPM 2.0 (Lomize et al., 2012) and the CHARMM-GUI Membrane Builder (Lee et al., 2019). The lipid bilayer membrane is comprised of DSM (upper leaf: 21.0%, inner leaf: 10.0%), POPC (35.0%, 15.0%), DOPC (3.5%, 1.5%), POPE (5.0%, 20.0%), DOPE (2.0%, 5.0%), POPS (0%, 15.0%), POPI (0%, 5.0%), POPA (2.2%, 0%), and CHOL (31.3%, 28.5%). The individually modeled lipids may be visualized in the CHARMM-GUI Archive—Individual Lipid Molecule Library (https://www.charmm-gui.org/?doc=archive&lib=lipid) (Jo et al., 2009).

To model the Na$_V$1.5/β1 trans interactions, the Na$_V$1.5 structure (PDB ID: 6QLA) inserted into the membrane described above was aligned to the Na$_V$1.4/β1 structure (PDB ID: 6AGF) and the linker region connecting Ig-domain to the transmembrane domain of β1 was reoriented toward an opposing membrane using Foldit Standalone (Kleffner et al., 2017). Protein–protein docking of the β1 Ig-domains was performed with ZDOCK (Pierce et al., 2014) and HADDOCK (Dominguez et al., 2003; van Zundert et al., 2016) using default settings to predict dimerization states.

### Molecular dynamics simulations

Conventional and steered all-atom molecular dynamics simulations were prepared and run using GROMACS 2021.3 (Abraham et al., 2015) implemented with the University of Cambridge High-Performance Computing resources. Periodic boundary conditions were established in a 150 × 150 × 150 Å cubic box. Each system was solvated in 150 mM NaCl with a zero net charge using the TIP3P model (Joung and Cheatham, 2008) and run at 310 K (Bondi, 1964). The particle-mesh Ewald method was used to calculate long-range electrostatic interactions, and the cutoff for Coulomb interactions and van der Waals interactions was set to 10 Å (Darden et al., 1993). The LINCS algorithm was used to constrain molecular bonds (Hess et al., 1997). All systems were subjected to steepest descent minimization, upon which six series of a 125 ps NPT equilibration ensemble with temperature coupling using velocity rescaling (Bussi et al., 2007) and pressure coupling using the Parrinello–Rahman method were conducted (Parrinello and Rahman, 1981). All simulations were run using the CHARMM36 all-atom force field (C36 FF) with 2 fs time steps (Huang et al., 2017a). Duplicate conventional productions of 150 ns each were run for the glycosylated Na$_V$1.4 and Na$_V$1.5. Steered molecular dynamics simulations were performed using constant velocity stretching by applying an external force to the M154 residue of both β1 subunits (Lu and Schulten, 1999). A virtual harmonic spring attached to the selected residues was pulled at a constant velocity of 5 nm/ns with a force constant of 100 kJ/mol nm$^2$ in opposite directions (Guzmán et al., 2008). One steered production replicate was performed for the side-to-side and tip-to-tip models each (Sheridan et al., 2019). VMD was used to visualize snapshot structures from the productions (Humphrey et al., 1996). Snapshot structures were extracted every 5 ns from the resultant trajectory files and overlapped onto the respective starting glycosylated membrane-bound Na$_V$1.4 and Na$_V$1.5 structures to represent glycan flexibility and conformational landscape over the course of the simulations (Figs. 4 and 5) (Pronk et al., 2013).

### Data availability

Trajectories of molecular dynamics simulations and structural models are available upon request due to data size.

## Acknowledgments

Jeanne M. Nerbonne served as editor.

We greatly thank the team at the University of Cambridge High Performance Computing Centre.

S.C. Salvage was supported by the British Heart Foundation (PG/19/59/34582 to S.C. Salvage, C.L.-H. Huang, and A.P. Jackson). S.J. Arundel was funded by the Biochemical Society Summer Vacation Studentship.

Author contributions: C.A. Beaudoin: Conceptualization, Data curation, Formal analysis, Investigation, Methodology, Resources, Software, Validation, Visualization, Writing - original draft, Writing - review & editing, M. Kohli: Conceptualization, Data curation, Formal analysis, Investigation, Methodology, Software, S.C. Salvage: Conceptualization, Supervision, Writing - review & editing, H. Liu: Conceptualization, Data curation, Writing - review & editing, S.J. Arundel: Conceptualization, Methodology, Writing - review & editing, S.W. Hamaia: Conceptualization, M. Lei: Resources, Writing - review & editing, C.L.-H. Huang: Conceptualization, Formal analysis, Funding acquisition, Methodology, Project administration, Writing - original draft, Writing - review & editing, A.P. Jackson: Conceptualization, Data curation, Formal analysis, Funding acquisition, Methodology, Project administration, Resources, Supervision, Validation, Writing - original draft, Writing - review & editing.

Disclosures: The authors declare no competing interests exist.

Submitted: 22 May 2024

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
