## [Peer Review File · The Journal of General Physiology]

Isoform-specific N-linked glycosylation of NaV channel α -subunits alters β -subunit binding sites

Christopher Beaudoin, Manas Kohli, Samantha Salvage, Hengrui Liu, Samuel Arundel, Samir Hamaia, Ming Lei, Christopher L.H. Huang, and Antony Jackson

Corresponding Author(s): Antony Jackson, University of Cambridge

Review Timeline:

Submission Date:	May 22, 2024
Editorial Decision:	June 25, 2024
Revision Received:	September 19, 2024
Editorial Decision:	September 20, 2024
Revision Received:	November 22, 2024

Editor: Jeanne Nerbonne

Transaction Report:

DOI: <https://doi.org/10.1085/jgp.202413609>

June 25, 2024

Dr. Antony P Jackson
University of Cambridge
Department of Biochemistry
Tennis Court Road
Hopkins Building
Cambridge CB2 1QW
United Kingdom

Re: 202413609

Dear Dr. Jackson,

Thank you for submitting your manuscript, titled "Isoform-specific N-linked glycosylation of voltage-gated sodium channel α -subunits alters β -subunit binding sites", to the Journal of General Physiology (JGP). Your manuscript has now been seen by three reviewers, whose comments are appended below. As you will see, the reviewers were very enthusiastic about the study and its potential impact. They have, however, raised several minor concerns that the editors agree should be addressed.

We hope that you will be able to submit a revised manuscript that addresses the points noted in a revised manuscript and/or in written reply. Please do not hesitate to contact me (via the editorial office) if you feel that a discussion of this decision letter and/or the reviewers' comments would be helpful.

Please submit your revised manuscript via the link below, along with a point-by-point letter that details your response to the reviewers' specific comments, as well as a copy of the text with alterations highlighted (boldfaced or underlined). If the article is eventually accepted, it would include a 'revised date' as well as submitted and accepted dates. If we do not receive the revised manuscript within one year, we will regard the article as having been withdrawn. We would be willing to receive a revision of the manuscript at a later time, but the manuscript will then be treated as a new submission, with a new manuscript number.

Please pay particular attention to recent changes to our instructions to authors in the following sections: Data presentation, Blinding and randomization and Statistical analysis, under Materials and Methods, as shown here: <https://rupress.org/jgp/pages/submission-guidelines#prepare>. Re-review will be contingent on inclusion of the required information (including for data added during revision) and demonstration of the experimental reproducibility of the results. Also, To improve the reproducibility of published content, we have partnered with SciScore. Authors are prompted in eJP to copy and paste the Materials and Methods section of their manuscript for a SciScore assessment when submitting their revised manuscript. Authors are encouraged (not required) to further revise their Materials and Methods if the SciScore is below 4. More information can be found here: <https://rupress.org/jgp/pages/submission-guidelines#sciscore>.

Please note, JGP now requires authors to submit Source Data used to generate figures containing gels and Western blots with all revised manuscripts (when applicable). This Source Data consists of fully uncropped and unprocessed images for each gel/blot displayed in the main and supplemental figures. If your paper includes cropped gel and/or blot images, please be sure to provide one Source Data file for each figure that contains gels and/or blots along with your revised manuscript files. File names for Source Data figures should be alphanumeric without any spaces or special characters (i.e., SourceDataF#, where F# refers to the associated main figure number or SourceDataFS# for those associated with Supplementary figures). The lanes of the gels/blots should be labeled as they are in the associated figure, the place where cropping was applied should be marked (with a box), and molecular weight/size standards should be labeled wherever possible. Source Data files will be made available to reviewers during evaluation of revised manuscripts and, if your paper is eventually published in JGP, the files will be directly linked to specific figures in the published article.

Source Data Figures should be provided as individual PDF files (one file per figure). Authors should endeavor to retain a minimum resolution of 300 dpi or pixels per inch. Please review our instructions for export from Photoshop, Illustrator, and PowerPoint here: <https://rupress.org/jgp/pages/submission-guidelines#revised>

Whilst you are revising your manuscript, we ask that you consider whether you have any artwork that might be suitable for the cover of JGP. Microscopy images are particularly good for cover artwork, but other types of image can be very effective, so we encourage you to be creative. Please don't restrict yourself to images from the paper; an image that is relevant to the work described would be just as suitable. Images should be a minimum resolution of 300 dpi. To see recent examples, visit the following page and click on 'Show covers? Yes': <https://jgp.rupress.org/content/by/year>

Thank you for submitting your interesting research to JGP.

Please submit your revised manuscript, and any associated files, via this link:
Link Not Available

Sincerely,

Jeanne Nerbonne, Ph.D.
On behalf of Journal of General Physiology

Journal of General Physiology's mission is to publish mechanistic and quantitative molecular and cellular physiology of the highest quality; to provide a best-in-class author experience; and to nurture future generations of independent researchers.

Reviewer #1 (Comments to the Authors):

The study utilizes cryo-EM models to provide a detailed structural analysis of Nav channels, identifying specific glycosylation sites and their potential functional impacts. This high-resolution technique offers robust data on the positioning and structural roles of glycans. The research connects structural findings with clinically relevant mutations, linking specific glycosylation site disruptions to pathologies such as Dravet Syndrome and Brugada Syndrome. This highlights the medical significance of the structural features being studied.

The text needs to be perforated for typos as "intermembrane" in line 317, or line 219 "different types of action propagation specific to the tissue." correct to "different types of action potential propagation specific to the tissue." Additionally, consistency in the use of the NaV, the V is capital and subscript.

Redundant citation: Line 93: "Specifically, human β 1, β 2, and β 4 have, respectively, been resolved with Nav1.1 (Pan et al., 2021), Nav1.3 (Li et al., 2022), Nav1.4 (Pan et al., 2018), Nav1.6 (Fan et al., 2023), and Nav1.7 (Shen et al., 2019; Huang et al., 2022); Nav1.3 (Li et al., 2022), Nav1.6 (Fan et al., 2023), and Nav1.7 (Shen et al., 2019; Huang et al., 2022); and Nav1.1 (Pan et al., 2021)."

Line 229, the phrase "Importantly, in both channels, the pore is not blocked by glycans.", could be reorganized as "In both channels, the pore is not blocked by glycans."

Further explanation for the phrase at line 478. "The β -subunits affect the electrophysiological properties of the Nav channel α -subunits." How does the β -subunit affect the electrophysiological properties of the NaV channel α -subunits?

At line 319: "Upon measuring the predicted distance between the lipid head groups, the tip-to-tip model shows a distance of 18.72 nm and the side-to-side model shows 15.34 nm." What do those distances mean? Mention the significance of these distances in the same paragraph.

At line 466: "The analysis also identified a N-linked glycan unique to Nav 1.8 which would occlude the binding sites of the β 2 and β 4 Ig-domains.", this information needs more context.

Reviewer #2 (Comments to the Authors):

The authors set out to begin to explore the structural impact of N-glycans attached to voltage-gated Na⁺ channel alpha subunits (Nav1-9). The analysis starts with Nav1-9 cryo-EM data that showed the first residue of N-glycans, N-acetylglucosamine, attached to extracellularly-facing Asparagine residues. A limitation of the available cryo-EM data is that the structure/location of the remainder of the heterogenous and typically hybrid and complex channel N-glycans are not resolved. Thus, herein, the authors modeled the structure and location of a single bi-antennary form of N-glycan structure with a single sialic acid attached at the terminal end of the N-glycan chain. The work is very interesting, important, and nicely extends the structural information provided by the alpha subunit Cryo-EM studies.

There are a significant set of biophysical data describing putative mechanisms by which Nav alpha and beta subunit N-glycans (particularly sialic acids) confer functional effects on Nav activity. Many of the reports that are most directly related to the structural data described in this manuscript, were published quite a few years ago and have not been further investigated due, in large part, to the significant limitation at the time of these prior works - a lack of N-glycan structural data. Thus, the data presented here provides a rare opportunity - the potential to overcome this major limitation of previous studies through interpretation of prior data in the context of these channel structures that now include predicted N-glycan structures. Conversely, because a single bi-antennary N-glycan structure was used to model N-glycan structure (of alpha subunits only), interpretation of the current data in context of these historical functional data could also serve to validate the current structural modeling described herein, thereby markedly increasing the significance and potential impact of the current study.

There is another limitation to the current hypothesis with respect to determining inter- and intra-subunit interactions that the authors should consider acknowledging. That is, the reliability of accurately predicting alpha-beta and beta-beta subunit interactions is limited in the current study because the N-glycan structures of only alpha subunits were modeled. Beta subunits are also heavily N-glycosylated, and beta subunit N-glycans were shown to confer function on alpha subunit gating. However, here, because only alpha subunit N-glycan structure/location were modeled, only a portion of the N-glycan structures potentially

involved in subunit interactions was used to predict inter- and intra-subunit interactions. While this reviewer is enthusiastic to see the analysis to include beta subunit N-glycan structure/location, a relatively straightforward discussion of this limitation would suffice if the authors prefer not to include yet another set of studies that might well be better suited for a subsequent report.

Finally, the organization of the paper in its current iteration, including the somewhat diffuse set of studies described, made it difficult to follow the apparent high impact of the study.

In summary, this manuscript provides a very important structural extension to the large body of Cryo-EM work on Nav alpha and alpha/beta subunit structures. The authors have a wonderful opportunity to further extend their structural work by adding more detailed interpretations of existing biophysical data that will likely provide additional insight into how the structure/location of N-glycans contribute to their the functional (biophysical) role on Nav activity. This additional discussion would not only improve the readability of the manuscript, but more importantly, the significance and impact of this work while likely overcoming limitations associated with the current study. One of the reviews (likely others) cited in the current manuscript discuss a relatively complete set of studies published prior to the review's 2012 publication date that describe data demonstrating functional roles for alpha and beta subunit N-glycans - *Compr Physiol.* 2012 Apr;2(2):1269-301. doi: 10.1002/cphy.c110044.

Below, please find an example of a set of existing functional data that might be interpreted in the context of the new structures: The structure/location of the exemplar N-glycan structure relative to the structure of each of the nine alpha subunits as reported already is really beautiful work (Figs. 3 and 4). However, one aspect of Figure 3 that was confounding to this reviewer was the lack of interpretation of prior data with respect to the unique and large set of N-glycans attached to ECTL 1 of Nav1.4. This unique set of N-glycan structures was shown many years ago to be responsible for most (all?) functional effects of Nav1.4 N-glycans (sialic acids) on Nav gating (JGP, Mar;109(3):327-43. doi: 10.1085/jgp.109.3.327; *J Physiol.* 2002 Feb 1;538(Pt 3):675-90. doi: 10.1113/jphysiol.2001.013285). A second report (*J Physiol.* 2002 Feb 1;538(Pt 3):675-90. doi: 10.1113/jphysiol.2001.013285) showed that chimeras in which ECTL 1 was swapped between Nav1.4 and Nav1.5, resulted in a phenotype switch. That is, the effects of sialic acids on each alpha subunit construct's gating (including chimeras) was determined by the ECTL 1 that was present - only the Nav1.4 ECTL1 conferred sialic acid-dependent gating effects. Thus, here, interpretation of these existing functional data (and other data, as relevant) in the context of the newly described Nav1.4 and Nav1.5 structures that each contain unique N-glycan structures/locations, should support both existing and new data sets. Such analysis would provide a powerful extension of the current work, increasing its impacts by demonstrating the role of N-glycan structure/location to Nav activity, while also potentially overcoming an acknowledged limitation of the current study - using a single N-glycan structure to model N-glycan structure/location.

Reviewer #3 (Comments to the Authors):

This is a very interesting and well written Hypothesis piece describing how Nav channel glycosylation status may influence beta subunit binding and functional activity of the adhesion domain. The work fits the remit of the JGP Hypothesis article and is conceptual in nature, but is well supported by experimental modelling evidence. The conclusions are thought-provoking and rightly warrant further research, particularly into how glycosylation status may influence channel function and cell-cell adhesion dependent on Nav complex composition. Overall, this manuscript is informative and well reasoned. I believe it is ready to publish in its current form without further modification.

Reviewer #1 (Comments to the Authors):

The study utilizes cryo-EM models to provide a detailed structural analysis of Nav channels, identifying specific glycosylation sites and their potential functional impacts. This high-resolution technique offers robust data on the positioning and structural roles of glycans. The research connects structural findings with clinically relevant mutations, linking specific glycosylation site disruptions to pathologies such as Dravet Syndrome and Brugada Syndrome. This highlights the medical significance of the structural features being studied.
Thank you very much for your kind comments and consideration of the manuscript.

The text needs to be perforated for typos as "intermembrane" in line 317, or line 219 "different types of action propagation specific to the tissue." correct to "different types of action potential propagation specific to the tissue."

Thank you very much for spotting these typos. Both are now corrected. Additionally, all mentions of "Ig domain / Ig-domain" have been changed exclusively to "Ig-domain".

Additionally, consistency in the use of the NaV, the V is capital and subscript.

Thank you very much for suggesting the canonical use of Na_v, which has now been implemented in all text and figures.

Redundant citation: Line 93: "Specifically, human β 1, β 2, and β 4 have, respectively, been resolved with Nav1.1 (Pan et al., 2021), Nav1.3 (Li et al., 2022), Nav1.4 (Pan et al., 2018), Nav1.6(Fan et al., 2023), and Nav1.7 (Shen et al., 2019; Huang et al., 2022); Nav1.3 (Li et al., 2022), Nav1.6(Fan et al., 2023), and Nav1.7 (Shen et al., 2019; Huang et al., 2022); and Nav1.1 (Pan et al., 2021)."

Thank you very much for pointing out the repeated use of the citations. Since two β -subunits may bind to one Na_v channel, citations were repeated each time. For example, Nav1.7 was resolved in complex with B1 and B2 in two publications: Shen et al., 2019 and Huang et al., 2022. Therefore, they are mentioned both for Na_v1.7/B1 and Na_v1.7/B2. We kindly request to keep the text as it is currently written to maintain thoroughness.

Line 229, the phrase "Importantly, in both channels, the pore is not blocked by glycans.", could be reorganized as "In both channels, the pore is not blocked by glycans."

Thank you for your comment. The word "importantly" has now been removed.

Further explanation for the phrase at line 478. "The β -subunits affect the electrophysiological properties of the Nav channel α -subunits." How does the β -subunit affect the electrophysiological properties of the NaV channel α -subunits?

Thank you very much for your question. The mechanisms behind the structural influence of β -subunits on the Nav channels are incompletely understood. The sentence below now follows the mentioned phrase.

“

Binding of the β -subunits modulates key Na_v channel conformational dynamics during an action potential cycle, including shifting the voltage ranges of activation and inactivation and altering the rate of recovery from inactivation (Yu et al., 2005; Cusdin et al., 2010; Cummins et al., 2001). These effects vary with different α - and β -subunit combinations.

“

At line 319: "Upon measuring the predicted distance between the lipid head groups, the tip-to-tip model shows a distance of 18.72 nm and the side-to-side model shows 15.34 nm." What do those distances mean? Mention the significance of these distances in the same paragraph.

Thank you very much for your key question and suggestion. These distances represent the width of the cardiomyocyte perinexal space. The text below can now be found following this sentence in the manuscript,

“

Both predicted intermembrane distances agree with the experimentally-derived 15-30 nm between membranes that form the cardiomyocyte perinexal space (Rhett et al., 2012; Veeraraghavan et al., 2018). These data corroborate with previously published data that suggest that trans interactions between β 1 Ig-domains may be responsible for establishing the structural restraints of the cardiomyocyte perinexus (Gourdie, 2019).

“

At line 466: "The analysis also identified a N-linked glycan unique to Nav 1.8 which would occlude the binding sites of the β 2 and β 4 Ig-domains.", this information needs more context. Thank you very much for your suggestion. We have now edited this sentence to form the two pasted below.

“

The analysis also identified an N-linked glycan unique to Nav1.8 on the ECTL of DII. Notably, the unique glycan at this site would extend over the area of VSD I, thus occluding the binding sites of the β 2 and β 4 Ig-domains.

“

Reviewer #2 (Comments to the Authors):

The authors set out to begin to explore the structural impact of N-glycans attached to voltage-gated Na⁺ channel alpha subunits (Nav1-9). The analysis starts with Nav1-9 cryo-EM data that showed the first residue of N-glycans, N-acetylglucosamine, attached to extracellularly-facing Asparagine residues. A limitation of the available cryo-EM data is that the structure/location of the remainder of the heterogeneous and typically hybrid and complex channel N-glycans are not resolved. Thus, herein, the authors modeled the structure and location of a single bi-antennary form of N-glycan structure with a single sialic acid attached at the terminal end of the N-glycan chain. The work is very interesting, important, and nicely extends the structural information provided by the alpha subunit Cryo-EM studies.

Thank you very much for your kind comments and review of the manuscript.

There are a significant set of biophysical data describing putative mechanisms by which Nav alpha and beta subunit N-glycans (particularly sialic acids) confer functional effects on Nav activity. Many of the reports that are most directly related to the structural data described in this manuscript, were published quite a few years ago and have not been further investigated due, in large part, to the significant limitation at the time of these prior works - a lack of N-glycan structural data. Thus, the data presented here provides a rare opportunity - the potential to overcome this major limitation of previous studies through interpretation of prior data in the context of these channel structures that now include predicted N-glycan structures. Conversely, because a single bi-antennary N-glycan structure was used to model N-glycan structure (of alpha subunits only), interpretation of the current data in context of these historical functional data could also serve to validate the current structural modeling

described herein, thereby markedly increasing the significance and potential impact of the current study.

Thank you very much for bringing these important points to attention. Please see our responses below.

There is another limitation to the current hypothesis with respect to determining inter- and intra-subunit interactions that the authors should consider acknowledging. That is, the reliability of accurately predicting alpha-beta and beta-beta subunit interactions is limited in the current study because the N-glycan structures of only alpha subunits were modeled. Beta subunits are also heavily N-glycosylated, and beta subunit N-glycans were shown to confer function on alpha subunit gating. However, here, because only alpha subunit N-glycan structure/location were modeled, only a portion of the N-glycan structures potentially involved in subunit interactions was used to predict inter- and intra-subunit interactions. While this reviewer is enthusiastic to see the analysis to include beta subunit N-glycan structure/location, a relatively straightforward discussion of this limitation would suffice if the authors prefer not to include yet another set of studies that might well be better suited for a subsequent report.

Thank you very much for these thought-provoking comments. We agree that the structural modelling and simulations of N-glycans on the surfaces of Nav channels provide a fresh perspective on the functional roles of the glycans. We are, indeed, planning to further explore the implications of N-linked glycans on the β -subunits in a subsequent report that focuses more the electrostatic role of the glycans in ion conductance. We agree that the Discussion section would benefit from more explanation of the potential functional affects of N-linked glycans. The text below is now included in the manuscript.

“

The β -subunits affect the electrophysiological properties of the Nav channel α -subunits (Namadurai et al., 2015). Binding of the β -subunits modulates key Nav channel conformational dynamics during an action potential cycle, thus shifting the voltage ranges of activation and inactivation and enhancing inactivation and recovery from inactivation rates (Yu et al., 2005; Cusdin et al., 2010; Cummins et al., 2001). The Ig-domains of the β -subunits also contain N-linked glycosylation sites. Notably, sialic acids on the Ig-domains have been found to shift Nav channel voltage gating parameters (Johnson et al., 2004). Intra- and intersubunit interactions between α - and β -subunit glycans may, thus, influence either the structural conformational dynamics or further contribute to changes in membrane potential via the additional negatively-charged sialic acid moieties on the β -subunit N-linked glycans. Hence, these data may have important implications for understanding Nav1.5 and Nav1.8 channel gating (Vijayaragavan et al., 2004; O'Malley and Isom, 2015).

“

Finally, the organization of the paper in its current iteration, including the somewhat diffuse set of studies described, made it difficult to follow the apparent high impact of the study.

Thank you very much for this note. We hope that the responses to the following suggestions help better emphasize the impact of the study.

In summary, this manuscript provides a very important structural extension to the large body of Cryo-EM work on Nav alpha and alpha/beta subunit structures. The authors have a wonderful opportunity to further extend their structural work by adding more detailed interpretations of existing biophysical data that will likely provide additional insight into how the structure/location of N-glycans contribute to their the functional (biophysical) role on Nav activity. This additional discussion would not only improve the readability of the manuscript, but more importantly, the significance and impact of this work while likely overcoming limitations associated with the current study. One of the reviews (likely others) cited in the

current manuscript discuss a relatively complete set of studies published prior to the review's 2012 publication date that describe data demonstrating functional roles for alpha and beta subunit N-glycans - *Compr Physiol.* 2012 Apr;2(2):1269-301. doi: 10.1002/cphy.c110044.

Below, please find an example of a set of existing functional data that might be interpreted in the context of the new structures:

The structure/location of the exemplar N-glycan structure relative to the structure of each of the nine alpha subunits as reported already is really beautiful work (Figs. 3 and 4). However, one aspect of Figure 3 that was confounding to this reviewer was the lack of interpretation of prior data with respect to the unique and large set of N-glycans attached to ECTL 1 of Nav1.4. This unique set of N-glycan structures was shown many years ago to be responsible for most (all?) functional effects of Nav1.4 N-glycans (sialic acids) on Nav gating (JGP, Mar;109(3):327-43. doi: 10.1085/jgp.109.3.327; *J Physiol.* 2002 Feb 1;538(Pt 3):675-90. doi: 10.1113/jphysiol.2001.013285). A second report (*J Physiol.* 2002 Feb 1;538(Pt 3):675-90. doi: 10.1113/jphysiol.2001.013285) showed that chimeras in which ECTL 1 was swapped between Nav1.4 and Nav1.5, resulted in a phenotype switch. That is, the effects of sialic acids on each alpha subunit construct's gating (including chimeras) was determined by the ECTL 1 that was present - only the Nav1.4 ECTL1 conferred sialic acid-dependent gating effects. Thus, here, interpretation of these existing functional data (and other data, as relevant) in the context of the newly described Nav1.4 and Nav1.5 structures that each contain unique N-glycan structures/locations, should support both existing and new data sets. Such analysis would provide a powerful extension of the current work, increasing its impact by demonstrating the role of N-glycan structure/location to Nav activity, while also potentially overcoming an acknowledged limitation of the current study - using a single N-glycan structure to model N-glycan structure/location.

Thank you very much for the supportive comments and for pointing out the published functional data that is highly relevant to the study. Although we were initially wary of including insights from functional data, since this study is primarily focused on the structural implications of the N-linked glycans on Nav channel surfaces, we agree that the above and other studies may be corroborated by the manuscript data. To elaborate further, the text below is now included in the Discussion section of the manuscript.

“

The ECTL I of Nav_v1.4 was found to be the most densely glycosylated region with seven N-linked glycan sites among all human Nav channels. Previous studies demonstrated that sialylated N-linked glycans on the Nav_v1.4 ECTL I uniquely affect channel gating (Bennett et al., 1997; Bennett, 2002; Ednie et al., 2015). Notably, patch-clamp electrophysiology of a chimera of Nav_v1.5 with the ECTL I of Nav_v1.4 resulted in the shifting of all voltage-dependent parameters with respect to wild-type Nav_v1.5, while replacing the ECTL I of Nav_v1.4 with that of Nav_v1.5 resulted in no gating changes under reduced sialylation (Bennett, 2002). Furthermore, changes in expression of sialyltransferases (both artificially and during development) and enzymatic cleavage of sialic acids have been shown to affect Nav channel gating (Stocker and Bennett, 2006; Ednie et al., 2013). These data highlight the significant effect of position, location, and extent of sialylation of N-linked glycans on Nav channel surfaces. Considering that Nav_v1.4 is specifically expressed in skeletal muscle cells, the packing of negatively-charged sialic acids onto one extracellular loop may lead to unique tissue-specific effects on action potential propagation. Also, the glycan cloud on the ECTL I extends near but not over the channel pore; therefore, sialic acids on the tips of the glycan trees may affect sodium ion conductance through the channel by contributing to the negative membrane potential or interacting with sodium ions directly (Bennett et al., 1997). Future work looking into the biophysical effects of these uniquely nested N-linked glycans on Nav_v1.4 may shed light on the roles of negatively-charged glycans on Nav and other voltage-gated ion channels.

“

Reviewer #3 (Comments to the Authors):

This is a very interesting and well written Hypothesis piece describing how Nav channel glycosylation status may influence beta subunit binding and functional activity of the adhesion domain. The work fits the remit of the JGP Hypothesis article and is conceptual in nature, but is well supported by experimental modelling evidence. The conclusions are thought-provoking and rightly warrant further research, particularly into how glycosylation status may influence channel function and cell-cell adhesion dependent on Nav complex composition. Overall, this manuscript is informative and well reasoned. I believe it is ready to publish in its current form without further modification.

Thank you very, very much for the kind words and review of the manuscript. We look forward to future responses from the scientific community responds to the reported insights and building upon these findings in our upcoming work.

September 23, 2024

Dr. Antony P Jackson
University of Cambridge
Department of Biochemistry
Tennis Court Road
Hopkins Building
Cambridge CB2 1QW
United Kingdom

Re: 202413609R1

Dear Dr. Jackson,

I am pleased to let you know that your manuscript, titled "Isoform-specific N-linked glycosylation of voltage-gated sodium channel α -subunits alters β -subunit binding sites" is scientifically acceptable for publication in Journal of General Physiology. Formal acceptance will follow when it is modified in accordance with the referees' remarks and our editorial policies.

Please note items that need attention are listed at the bottom of this email (under 'manuscript formatting checklist') and on the attached marked-up pdf file. Please also be sure to include a letter addressing the reviewers' comments point-by-point (if applicable) and a copy of the text with alterations highlighted (boldfaced or underlined). Your manuscript should be a double-spaced MS Word file and include editable tables, if appropriate.

JGP requires a data availability statement for all research article submissions. These statements will be published in the article directly above the Acknowledgments. The statement should address all data underlying the research presented in the manuscript. Please visit the JGP instructions for authors for guidelines and examples of statements at <https://rupress.org/jgp/pages/editorial-policies#data-availability-statement>.

Lastly, JGP adds short captions to articles listed on our weekly newest article emails. If you haven't, please provide a short, ~40-word summary statement for the online JGP table of contents and alerts. This summary should describe the context and significance of the findings for a general readership and be placed on/near the title page.

Please submit your final files via this link:
Link Not Available

Thank you for choosing to publish your research in JGP and please feel free to contact me with any questions.

Sincerely,

Jeanne Nerbonne, Ph.D.
On behalf of Journal of General Physiology

Journal of General Physiology's mission is to publish mechanistic and quantitative molecular and cellular physiology of the highest quality; to provide a best in class author experience; and to nurture future generations of independent researchers.

Manuscript formatting checklist:

- MS Word document of text needed (including editable tables)
- MS Word document of supplemental text needed, if applicable (including figure legends and editable tables)
- Brief Statement describing supplementary information needed, if applicable (in subsection at end of Materials & Methods)
- Please include a data availability statement preceding the Acknowledgments section. Please see <https://rupress.org/jgp/pages/editorial-policies#data-availability-statement>
- Figures created at sufficient resolution and in acceptable format (including supplemental if applicable). If working in Illustrator, we prefer .ai or .eps file format. If working in Photoshop please use 600dpi/1000dpi .tiff or .psd file format. Minimum resolution at estimated print size: Minimum resolution for all figures is 600 dpi. For figures that contain both photographs and line art or text, 600 dpi is highly recommended. Figures containing only black and white elements (line art, no color, and no gray) should be 1,000 dpi. Maximum figure size is 7 in wide x 9 in high (17.5 x 22.8 cm) at the correct resolution. <https://jgp.rupress.org/fig-vid-guidelines>
- Supplemental figures, if any, conforming to same guidelines as manuscript figures (noted above)
- If images resemble one from a prior publications, the author must seek permissions (to reproduce or adapt) from the original

publisher. [You can resubmit your paper while waiting to hear back from the original publisher but please keep us updated]
- All authors must complete a disclosure form prior to acceptance. A link to complete the form has been sent to all coauthors.
Please provide the editorial office with updated email addresses if necessary